# Eureka-Moments in Transformers: Multi-Step Tasks Reveal Softmax Induced Optimization Problems

## Abstract

In this work, we study rapid, step-wise improvements of the loss in transformers when being confronted with multi-step decision tasks. We found that transformers struggle to learn the intermediate tasks, whereas CNNs have no such issue on the tasks we studied. When transformers learn the intermediate task, they do this rapidly and unexpectedly after both training and validation loss saturated for hundreds of epochs. We call these rapid improvements Eureka-moments, since the transformer appears to suddenly learn a previously incomprehensible task. Similar leaps in performance have become known as Grokking. In contrast to Grokking, for Eureka-moments, both the validation *and* the training loss saturate before rapidly improving. We trace the problem back to the Softmax function in the self-attention block of transformers and show ways to alleviate the problem. These fixes improve training speed. The improved models reach 95% of the baseline model in just 20% of training steps while having a much higher likelihood to learn the intermediate task, lead to higher final accuracy and are more robust to hyper-parameters.

## 1 Introduction

A key quality of any intelligent system is its ability to decompose a problem into sub-problems and learn to solve these sub-problems even in the absence of direct feedback. Deep learning has been enabling such capabilities to a certain degree. For example, deep classifiers learn the hierarchical feature representations necessary to build a good classifier. Language models build groups of tokens required to derive the meaning of a token and hence the whole sentence. Reinforcement learning learns object representations required to predict how to receive a sparse reward. While aforementioned examples show great promise, there is large effort that researchers or practitioners spend in designing the training process to learn sub-tasks. For instance, reward shaping is a common practice in reinforcement learning and many computer vision works propose explicit or implicit intermediate supervision. At the same time, we are not aware of a study on implicit multi-step learning.

One way to find out more about it could be to study popular tasks, for which many benchmarks and results already exist, in more detail. In fine-grained classification, the network could initially learn super-classes and subsequently refine its classification. In BERT pretraining (Kenton & Toutanova, 2019) the network might first learn word frequencies and only later learns to use the context. Indeed, RoBERTa seems to learn these tasks consecutively and improves on the second task rapidly after an early saturation phase (Fig. 1c). However, real data prohibits a clean study: 1) The exact sub-tasks are typically unknown and, hence, are hard to study. 2) Not all samples may require multi-step reasoning and there could be many learning shortcuts (Geirhos et al., 2020), making progress on the multi-step task unrecognizable. 3) The features necessary for the tasks are unknown, i.e., we cannot study what the network fails to learn, let alone the reason for it. 4) Even the number of steps is unknown, thus, we cannot determine if models learn only a subset of the tasks.

As a remedy, we propose a new paradigm to analyse learning of multi-step tasks: we fully control the data-generating process through synthetic data generation. This allows us to synthetically create various two-step tasks in a controlled setting that facilitates a detailed study of such. Through the synthetic tasks we can remove confounding variables, know the number of tasks, know for each task

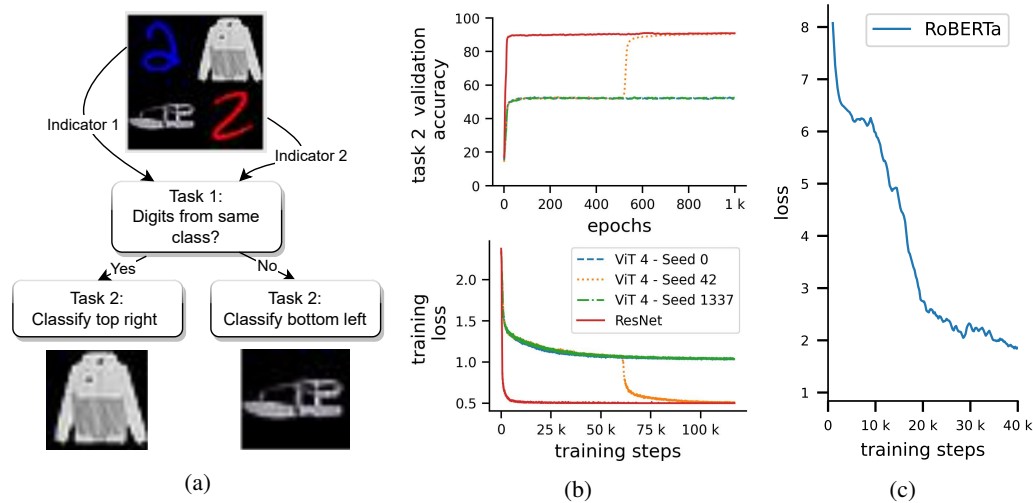

Figure 1: **Transformers can get stuck during optimization for two-stage tasks. (a)** Describes our 2 step decision task used to study Eureka-moments. **Task 1** is to compare the two **indicators** (here digits). If the digits are the same, **task 2** is to classify the top-right image and bottom left else. Top-right and bottom left are referred to as **targets**. The location of the correct target is referred to as **target location**. **(b) Validation accuracy** and **training loss** for the task in (a). 2 ViTs (blue and green) fail to converge, while ViT with seed 42 (yellow) has an Eureka-moment. Eureka-moments are characterised by a sudden increase of accuracy and drop of the loss. ResNets are not susceptible to this kind of optimization difficulty. **(c) Eureka-moments for a real dataset.** Sharp drops of the training loss after initial plateauing can also be observed for RoBERTa pretraining.

the relevant features, their location and the total number of steps. Thus, it solves the issues above all at once. This comes with the explicit assumption, that findings on synthetic data transfer to real data. We provide some indications, that this assumption holds. In each of our datasets, the answer to the first task $p(z|x)$, which is not explicitly represented in the loss, must be found by the model in order to correctly solve the final second task depending on it $p(y|x, z)$. For instance, in Fig. 1a the model must first classify the two digits to find out if they are the same class, which determines where to look for the subsequent fashion classification task. The loss only provides a training signal about the latter classification task, not whether it solved the first task correctly. Thus, the model must figure this out by itself during training. Formally, the task can be described as $p(y|x, z) \cdot p(z|x)$, i.e. the probability of class $y$ given evidence $x$ and the latent variable $z$.

Our study reveals that transformers have significant difficulties in learning such simple two-step tasks (Fig. 1b). After they learned to classify the fashion images $p(y|x, z)$, they just saturate and after a long time suddenly learn $p(z|x)$, i.e. the task to compare the digits. We call this phenomenon a Eureka-moment. With other initializations they never learn the taks 1 within 1000 epochs and stick with the prior $p(z)$. The probability of Eureka-moments depends on the difficulty of the task, as we will reveal later. In contrast, a ResNet learns both tasks immediately.

Our goal is not to add to the widespread transformer vs. CNN discussion, but we want to investigate this particular problem. What is its cause? Is it due to a too small capacity? Is it the number of heads or the learning rate? Is it the arrangement of digits and fashion images? We found that these factors play a minor role and finally traced the problem back to the Softmax function in the transformer's attention blocks. We found parts of the gradient's components become very small depending on the Softmax's output. This cannot be fixed by simply using a larger learning rate, since other components of the gradient are large, but it can be fixed by a simple normalization of the Softmax function.

In summary, the contributions of this analysis paper are: 1) we study multi-step learning without intermediate supervision by fully controlled data-generating process through synthetic data generation. 2) We discover a new failure mode of transformers. 3) We analyse the mechanisms underlying this failure mode and find that the Softmax function causes small gradients for the key and value weight matrics, thereby hindering learning. 4) To validate the role of Softmax, we show to mitigate the failure mode through small interventions. We show that these interventions lead to significantly faster

convergence, higher accuracy, higher robustness towards hyper-parameters and higher probability of model convergence, affirming our analysis. Code will be made available upon acceptance.

## 2 RELATED WORKS

**Emergence and phase transitions.** Our work is weakly connected to work on emergence and phase transitions as defined by Steinhardt (2022); Wei et al. (2022). Here, *emergence* refers to a qualitative change in a system resulting from a quantitative increase of either model size, training data or training steps. *Phase transitions* describe the same phenomenon but require the change to be sharp. Eureka-moments are special types of phase transitions. Recent discoveries of phase transitions might be an artefact of the discontinuous metrics (Schaeffer et al., 2023; Srivastava et al., 2022). In our case, also smooth metrics change rapidly. A connection between phase transitions or emergence and our work may exist, and both may be related to escaping bad energy landscapes as in our work.

**Unexplained phase transitions.** Previous works reported observations that may be Eureka-moments, without investigating their cause. For instance, rapid improvements happen for in-context-learning (Olsson et al., 2022) and BERT training (Gupta & Berant, 2020; Deshpande & Narasimhan, 2020; Nagatsuka et al., 2021). Deshpande & Narasimhan (2020) propose to bias attention towards predefined patterns and observed speed-up in BERT training. In our work, we also identify the learning of the right attention pattern to be the problem, suggesting a common underlying phenomenon.

**Grokking.** A similar phenomenon has been discovered on synthetic data (Power et al., 2022) and was further studied in (Liu et al., 2022b; Nanda et al., 2023; Thilak et al., 2022; Millidge, 2022; Barak et al., 2022; Liu et al., 2022a). Grokking describes the phenomenon of sudden generalization after overfitting, which can be induced by weight decay. In contrast to Eureka-moments, the training accuracy already saturates at close to $100\,\%$ (overfitting), a long time before the validation accuracy has a sudden leap from chance level to perfect generalization. For Eureka-moments, validation and training loss saturate (no overfitting) and the sudden leap occurs for both simultaneously.

**Unstable gradients in transformers.** The position of the layer-norm (LN) (Xiong et al., 2020) and instabilities in the Adam optimizer in combination with LN induced vanishing gradients (Huang et al., 2020). Removing the LN (Baevski & Auli, 2018; Child et al., 2019; Wang et al., 2019) or Warmup (Baevski & Auli, 2018; Child et al., 2019; Wang et al., 2019; Huang et al., 2020) resolves this problem, but in our case, Warmup alone does not help. Others identified the Softmax as one of the problems, showing that both extremes, attention entropy collapse, i.e. too centralized attention (Zhai et al., 2023; Shen et al., 2023) and a large number of small attention scores, i.e. close to maximum entropy (Dong et al., 2021; Chen et al., 2023) can lead to small gradients (Noci et al., 2022). As a remedy to vanishing gradients caused by entropy collapse Wang et al. (2021) proposed to replace the Softmax by periodic functions. However, before Eureka-moments, the attention distribution is in neither extreme. Instead the attention is allocated to the wrong tokens.

**Temperature in Softmax.** A key operation in the transformer is the scaled dot-product attention. Large products can lead to attention entropy collapse (Zhai et al., 2023; Shen et al., 2023), which results in very small gradients. In contrast, Chen et al. (2023) observed close to uniform attention over tokens. They scaled down very low scores further while amplifying larger scores, but this only amplified the problem when important tokens are already ignored. Instead, Jiang et al. (2022) proposed to normalize the dot product. Their proposed *NormSoftmax* avoids low variance attention weights and thus avoids the small gradient problem. We found it as the most effective intervention on the Softmax function. Others proposed to learn the temperature parameter (Dufter et al., 2020; Ali et al., 2021), but this is difficult to optimize. For very large models the problem becomes more severe. Models with more than 8B parameters show attention entropy collapse (Dehghani et al., 2023). They followed Gilmer et al. (2023) and normalized the $QK^T$ with layer norm before the Softmax.

## 3 BACKGROUND

**Preliminaries.** This work investigates the dot product attention (Vaswani et al., 2017), defined as

$$\text{Attention}(Q, K, V) = \text{softmax}\left(\frac{QK^T}{\tau}\right) V, \tag{1}$$

where the weight matrices $W_Q$, $W_K$ and $W_V$ map the input $X$ to query $Q$, key $K$, value $V$, and the temperature parameter $\tau$ controls the entropy of the Softmax distribution. A low temperature leads to low entropy, i.e. a more "peaky" distribution. Commonly, $\tau$ is set to $\sqrt{d_k}$, where $d_k$ is the dimensionality of $Q$ and $K$. Thus, $\sqrt{d_k}$ is the standard deviation of $QK^T$ under the independence assumption of rows of $Q$ and $K$ with 0 mean and variance of 1 (Vaswani et al., 2017).

**Softmax attention can cause vanishing gradients.** Attention entropy collapse, i.e. too centralized attention, can cause vanishing gradients (Zhai et al., 2023; Shen et al., 2023), since all entries of the Jacobian of the Softmax will become close to 0 (see appendix). Similarly, uniform attention can result in vanishing gradients for $W_K$ and $W_Q$, as shown by Noci et al. (2022).

A remedy to both problems is to control the attention temperature $\tau$. A larger $\tau$ in the Softmax will dampen differences of $QK^T$ and by that prevent vanishing gradients by low attention entropy. In contrast, a smaller $\tau$ will amplify differences of $QK^T$ and prevents vanishing gradients caused by uniform attention. Obviously, choosing the right temperature is difficult and can have a strong influence on what the model will learn, how fast it will converge etc. One of our interventions to test whether the Softmax is indeed the root cause is **Heat Treatment** (HT). We suggest to start training with a low temperature and follow a schedule to heat it up to the default value of $\sqrt{d_k}$. This approach has multiple advantages. First, the network gets optimized for "more peaky" attention, but the temperature increases steadily. By that, the network starts with centralized attention but since the next epochs attention will be more uniform than the previous, it does not run into the issue of low attention entropy. Second, the network can focus on most important features early in training and broaden the view over time, attending to other features.

**NormSoftmax.** An alternative to tame the attention is NormSoftmax (Jiang et al., 2022). Here, the expected standard deviation $\sqrt{d_k}$ gets replaced by the empirical standard deviation $\sigma(QK^T)$. This is done for each attention block in the transformers individually. NormSoftmax can be computed by

$$\text{NormSoftmax}(Q, K) = \text{Softmax}\left(\frac{QK^T}{min(\sigma(QK^T), \tau)}\right). \tag{2}$$

If $QK^T$ has low standard deviation differences will be amplified. If $\sigma(QK^T) > \tau$, $\tau$ will be used.

## 4 TASK DESCRIPTION AND EXPERIMENTAL CONDITIONS

**Task description.** All our datasets follow a multi-step structure. While arbitrarily many-step tasks are possible, we only consider two-step tasks in this work, as we found it sufficient to study the phenomenon. A schematic for one of our tasks can be seen in Fig 1a. The successive tasks in this work are referred to as **task 1** and **task 2**. Task 1 indicates what task 2 needs to do for a particular sample (e.g. where to look). For our vision datasets, the information provided by task 1 is which of the 2 possible targets (i.e. top or bottom) should be classified. On the vision datasets, task 2 is a simple classification task (what is the class at the indicated position?). Only task 2 is evaluated and, thus, direct supervision is only provided for task 2. By design of the datasets, $40-55\%$ accuracy can be obtained by only solving task 2 and picking the target at random. The range is due to varying difficulty of task 2. To obtain higher accuracy, both tasks must be solved.

**Vision dataset creation.** The visual datasets are based on MNIST (LeCun et al., 2010) and Fashion-MNIST (Xiao et al., 2017). An example and a schematic of the task is shown in Fig. 1a. For this dataset, samples are created by sampling 2 random Fashion-MNIST samples and 2 digit samples from the MNIST classes "1" and "2". We apply a random color to the MNIST samples (red or blue). Next, we compose a new image from the 4 images, putting the 2 MNIST samples on top left and bottom right and the Fashion-MNIST samples on the remaining free quadrants. If the 2 MNIST samples are from the same class, the class of the top-right image is the sample label and bottom-left else.

**Reasoning task.** To show that the problem is more general, we simplify the task to a minimum and create a **reasoning task**. We consider equations of the form: $f(a,\ b,\ c,\ d) = \begin{cases} c, & g(a,\ b) \\ d, & \text{otherwise} \end{cases}$,

where $a, b, c, d \in \{0,\ 1,\ ...,\ n\}$ and define task 1 as follows: $g(a,\ b) = \mathbb{1}[(\text{a is even} \wedge \text{b is odd}) \vee (\text{a is odd} \wedge \text{b is even})]$. Task 2 is just classification of $c$ or $d$. More details provided in A.12.

**Metrics.** We define the **Eureka-ratio** as the proportion of training runs with Eureka-moment across the different random seeds. To automatically detect Eureka-moments, we set a conservative threshold

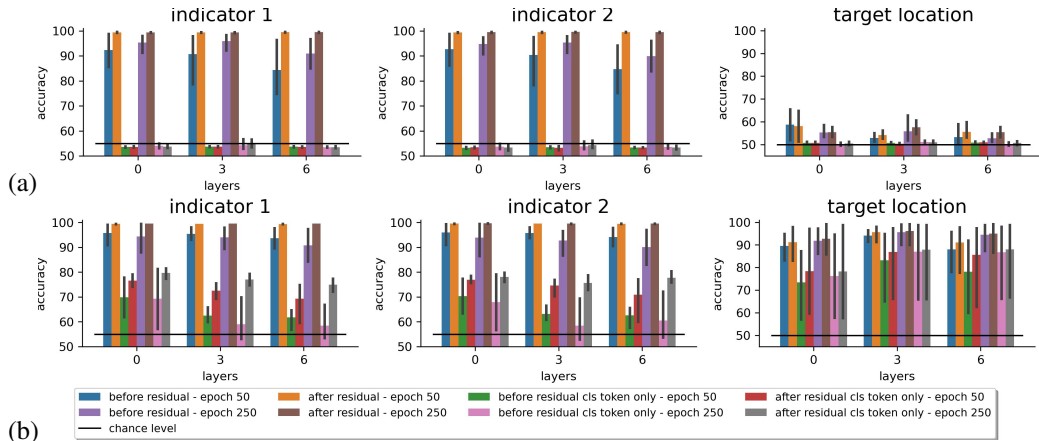

(a)

(b)

Figure 2: **Which information is represented in different parts of the network for (a) ViT (without Eureka-moment) and (b) ViT+HT (with Eureka-moment).** Bar plots show linear probe accuracies averaged over heads. Indicator 1 is the top MNIST digit, indicator 2 the other. Both, ViT and ViT+HT extract the indicator class information from the images and it is available in each layer. Information is available before and after the residual connection, therefore it is not entirely ignored by the attention. Differences between ViT and ViT+HT visible for CLS token and target location task. Error bars show variance over heads. More layers, representations and targets are shown in appendix.

at the validation accuracy of $70\%$, where $50\%$ can be reached solving only task 2. **Accuracy after Eureka-moment**, in the following referred to as accuracy, is computed over all runs that had an Eureka-moment. Thus, this metric has to be considered in conjunction with the Eureka-ratio, since high accuracy with low Eureka-ratio indicates a difficult optimization or a "lucky" run. Finally, **average Eureka-epoch** highlights the average epoch at which the Eureka-moment happened. It is computed only for runs with Eureka-moment and should be interpreted jointly with Eureka-ratio.

**Models and hyper-parameters.** Following Hassani et al. (2021), we use a ViT version specifically designed for small datasets like MNIST (LeCun et al., 2010) or CIFAR (Krizhevsky et al., 2009). Unless otherwise stated, we train a ViT with 7 layers, 4 heads and set the MLP-ratio to 2, a patch size of 4 and embedding dimension of 64 per head. As a result the default temperature is $\sqrt{d_k} = 8$. The ResNet we compared to has a comparable parameter count and consists of 9 layers. For ViT, ViT+HT and NormSoftmax we tested 5 different temperature parameters in initial experiments. More details on the architectures and training setup are provided in the appendix. For the reasoning task, we train a transformer on $30\%$ of the entire set of possible inputs (i.e. $11^4 = 14\,641$ input combinations) for 10K epochs over five random seeds. More details on model and training are provided in the appendix.

## 5 UNDERSTANDING EUREKA-MOMENTS AND THE OPTIMIZATION CHALLENGES OF TRANSFORMERS

Here, we analyze the problem on the dataset described in Fig. 1a. In Sec. 5.3 we provide experiments on 2 more datasts and finally show indications, that the results can be transferred to real datasets.

### 5.1 WHY DO TRANSFORMERS FAIL TO LEARN SIMPLE TWO-STEP TASKS?

To investigate why ViT's learning fails, we analyze the learned representations. Note that solving task 1 requires ViT to **1)** learn to distinguish the indicators, **2)** carry the information through the layers and **3)** compare the indicator information to obtain the target location. We investigate these questions in the following by learning linear probes on the output of the attention heads, i.e. $Z_i = \text{Attention}(Q_i, K_i, V_i)$, for all heads $i$. However, note that linear separation becomes very likely as the dimensionality of the features increase. Therefore, a high accuracy on the linear probe classification does not imply that the transformer is capable of using the information, only that it is represented and linearly separable. While the linear probe training has supervision on task 1, the transformer does not.

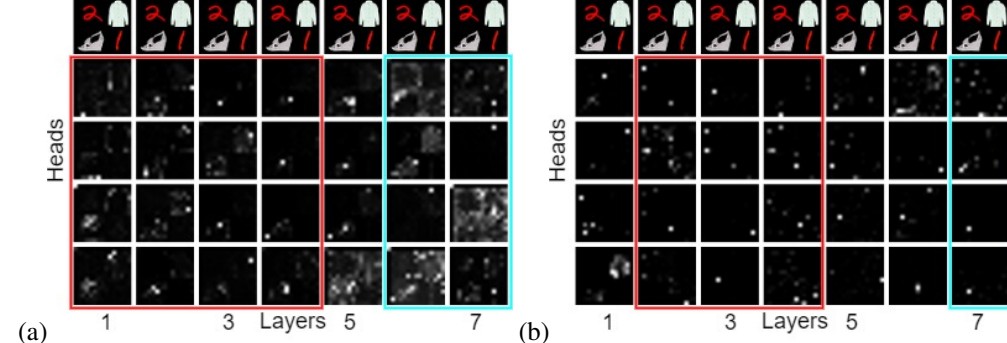

Figure 4: **Attention maps after training for: (a) ViT without Eureka-moment**, i.e. fails to compare the 2 digits. First layers explicitly ignore indicators (digits) (highlighted with red). **(b) ViT+HT with Eureka-moment** attends indicators in first layers (red) and predominantly attends the correct target (ankle boot) in later layers. Black is no attention and white is high attention. Maps show the average attention of each query $q$, i.e. we average over the key-dimension of the attention map.

**Does the transformer fail to distinguish the indicators?** The two left bar plots in Fig. 2a show that the indicators can be almost perfectly separated by linear probes in all layers. Still ViT fails to learn task 1 (orange, brown). Interestingly, the indicators are already separable in early stages of the training (orange). Consequently, the transformer represents the necessary features to possibly distinguish the indicators.

**Does the transformer filter out information required for task 1?** Since the loss provides a training signal only for task 2, the transformer may learn to just ignore the indicators. The residual connection in the transformer block makes ignorance of features unlikely but not impossible. Task 1 information could be ignored in the attention blocks s.t. it cannot attend to compare the indicator classes. To test this, we probe the representation after the attention operation before and after the residual connection. Fig. 2a reveals that the indicator information is available in all layers. Early in training, some indicator information is filtered out in the attention block (blue). Indicator information is partially filtered out in deeper layers, but is always recovered by the residual connections. We show the linear probe results based on the output of the Softmax $Z_i$. Plots for $Q_i$ and $K_i$ look similar. Therefore, it is evident that the features to solve task 1 are not filtered out.

**Does the transformer fail to combine the information?** We observe that the target location ( solution of task 1) cannot be inferred by the linear probe with high accuracy (Fig. 2a). This implies that even though the (indicator) information is present within the feature representations, it does not allow easy combination to predict the target location needed for task 2. In conclusion, the transformer has all the necessary information, but fails to combine it to solve the multi-step task.

**Differences to a transformer that had an Eureka-moment.** Fig. 2b shows the linear probe results for a transformer that learns task 1. Interestingly, the target location can be predicted by linear probes from all layers and all tested representations with high accuracy. This is in stark contrast to transformers that had no Eureka-moment. The second striking difference is that indicator information is represented in the CLS token. This is even stronger for early layers. We suspect that the information is written to the CLS token to solve the indicator class matching.

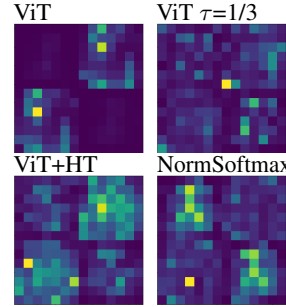

Figure 3: **Gradients on image for** $W_k$ at Epoch 50. For ViT the gradient for $W_K$ comes mostly from target regions, while for the other approaches indicator regions provide substantial gradient. Detailed explanation of this plot and plots for $Q$ and $V$ can be found in Sec. A.2

**How does the transformer fail to combine the information?** To obtain a comprehensive understanding of the reasons of the failure to combine information, we visualize the attention maps of two fully trained ViTs in Fig. 4. We find that the transformer without Eureka-moment does not attend the indicator digits and attends only the targets of task 2, whereas a transformer with Eureka-moment attends the indicator digits in early layers. This suggest that a ViT without Eureka-moment does not attend indicators enough to match them and has difficulties to learn to attend different regions.

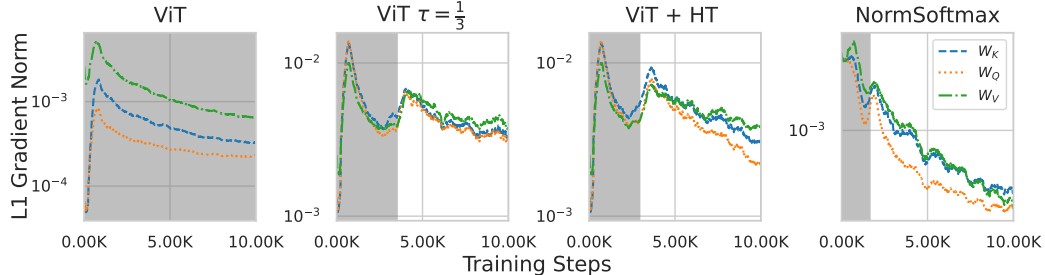

Figure 5: **L1 gradient norm during training** for $W_K$, $W_Q$ and $W_V$ for the first layer. For ViT, $W_K$ and $W_Q$ receive much smaller gradients than $W_V$. In particular before Eureka-moment (gray regions), the differences between gradient magnitudes are much smaller for smaller temperatures or NormSoftmax. The y-axis is log scaled. All layers shown in Fig. 12.

**Why does the transformer fail to learn to attend to the indicators?** Based on the discussion in Sec. 3, we suspect that ill-distributed attention scores lead to small gradients for $W_K$ and $W_Q$, which in turn inhibit learning. Note that high attention to some pairs with low attention to all others or uniformly distributed attention can result in vanishing gradients (Noci et al., 2022). To test this, we visualize the L1-norm of the gradient for the first layer in Fig. 5. For vanilla ViT that gradients for $W_K$ and $W_Q$ are 0.5-1 orders of magnitude smaller than those for $W_V$. Thus, small gradients are passing the Softmax, reaching $W_Q$ and $W_K$ and the attention map improves only slowly, which results in the observed learning difficulties. Differences between the gradients are much smaller for ViT $\tau = \frac{1}{3}$, ViT+HT and NormSoftmax, in particular before the Eureka-moment. Fig. 3 shows the origin of gradients on the image plane for $W_k$. For ViT, the gradients mostly originate from the target regions, which further explains why many steps are needed to move attention to the indicators.

**Is too small or too large attention-entropy the problem?** As discussed, low or high attention entropy can both result in vanishing gradients. We visualize the distribution of attention maps over training in Fig. 8. It is apparent that the vast majority of attention scores is very small, indicating that a too uniform attention is causing small gradients. Larger attentions are rare, but not absent, as can be seen in Fig. 4a, but indicator regions have small and uniform attention. Thus, we conclude that **local uniform attention** causes the transformer's learning problems.

## 5.2 CAN ENFORCING LOWER ENTROPY ATTENTION MAPS RESOLVE THE SMALL GRADIENTS?

In the previous subsection, we concluded that a local uniform attention is causing learning problems. To demonstrate this is indeed the case, we aim to modify the attention to remedy this issue. Particularly, we modify Softmax's temperature $\tau$ in the attention block. Large temperatures increase entropy, while small temperatures decrease it. We test various settings, training simply with lower temperature, HT, where the temperature increases from a low value to default temperature during the first half of the training and NormSoftmax, which adaptively changes the temperature for each sample, head and layer.

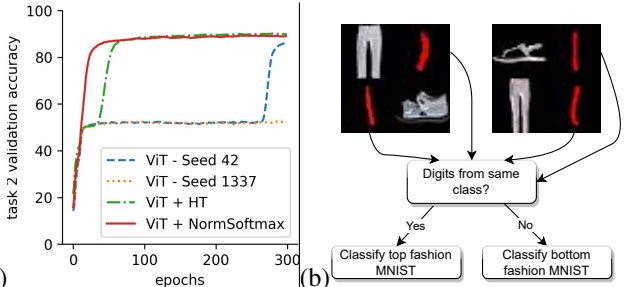

(a) (b)

Figure 6: **(a) Validation accuracy curves on main dataset. (b) No position task description.** This task removes more information by shuffling the rows of each data sample, i.e. classify either the top or bottom fashion sample. Two samples are shown to highlight differences from the task described in Fig. 1a.

**Does a lower temperature solve the small gradient issue and thereby mitigate the optimization issues?** Increasing the temperature from low to default or using NormSoftmax increases high attention scores (c.f. Fig. 8). Importantly, the transformer learns to also attend to the indicators (Fig. 4b). As a result, all approaches (lower temperature $\tau$, HT and NormSoftmax) solve the imbalanced gradient issue for $W_V$, $W_Q$ and $W_K$ (Fig 5) and lead to higher gradients in indicator regions (Fig. 3).

Table 1: **Comparison of proposed solutions and baselines.** For the **main dataset**, as described in Fig. 1a. and the **No position task** (Fig. 6b). $\tau$ not optimized for No position task. **ER**: Eureka-ratio, **Acc.**: Accuracy, **Avg. EE.**: average Eureka-epoch.

| | | Main Dataset | | | No Position Task | | |
|---|---|---|---|---|---|---|---|
| | | | Avg. over EMs | | | Avg. over EMs | |
| Model | $\tau$ | ER ↑ | Acc. ↑ | Avg. EE. | ER ↑ | Acc. ↑ | Avg. EE. |
| ViT | $\frac{1}{0.025}$ | 3/10 | 89.40 | 174.67 | - | - | - |
| ViT | $\frac{1}{0.075}$ | 6/10 | 90.13 | 181.34 | - | - | - |
| ViT + WD 0.5 | $\sqrt{d_k}$ | 5/10 | 90.09 | 177.8 | - | - | - |
| ViT | $\sqrt{d_k}$ | 7/10 | 89.48 | 207.43 | 0/4 | - | - |
| ViT + Warmup 20 | $\sqrt{d_k}$ | 8/10 | 87.65 | 205.87 | 1/4 | 89.55 | 117 |
| $W_{QKV}$ grad scaling | $\sqrt{d_k}$ | **10/10** | 87.96 | 119.4 | 0/4 | - | - |
| NormSoftmax | $\sqrt{d_k}$ | **10/10** | 89.56 | 28.2 | **3/4** | **88.98** | 228 |
| NormSoftmax | $\frac{1}{3}$ | **10/10** | 89.18 | 23.5 | 1/4 | 89.77 | 20 |
| ViT | $\frac{1}{3}$ | **10/10** | 89.35 | 66.6 | 1/4 | 89.68 | 191 |
| ViT+HT | $\frac{1}{3} \to \sqrt{d_k}$ | **10/10** | 89.81 | 74.0 | 1/4 | 88.36 | 242 |
| NormSoftmax + HT | $\frac{1}{3} \to \sqrt{d_k}$ | **10/10** | **89.83** | 17.5 | 1/4 | 90.63 | **19** |

Figure 7: **(a) Eureka-moments for single-layer transformers on a simple reasoning task.** We show the train (blue) and test (orange) accuracies for attention with Softmax, Softmax+HT, or NormSoftmax, over 5 random seeds (transparencies). Chance probability is $6/11 \approx 54\,\%$ (black). **(b) Eureka-moments in real dataset.** Sharp drops after initial plateauing can be observed also for RoBERTa pretraining. Using NormSoftmax leads to an earlier Eureka-moment.

As a result, the interventions indeed mitigate the optimization issues. Eureka-moments happen much earlier or instantly (see Fig. 6a). A comprehensive comparison between a vanilla ViT and other versions is provided in Tab. 1. Decreasing the temperature and using NormSoftmax increases the Eureka-ratio, accuracy and decreases the Eureka-epoch (i.e. improving the energy landscape). In contrast, increasing the temperature has a negative effect on the Eureka-ratio, supporting that the local uniform attention is the main cause for the learning problem.

### 5.3 IS THIS AN ARTIFICIAL PROBLEM CAUSED BY OTHER FACTORS?

**Does the transformer simply ignore specific indicator locations?** The task and dataset used so in the previous subsetctions showed indicators and targets always at the same location, i.e. indicators on top-left and bottom right. Such a dataset design might result in two undesired effects: 1) The transformer might learn to ignore features based on the associated positional embeddings. 2) The task might be easier, since positional embeddings can be used as shortcut to find indicators without the need to rely on the actual features. To test for both cases we create another dataset. Refer to Fig. 6b for the dataset and task description. We observe that removing the fixed position for indicators and targets makes the task even more difficult (Tab. 1 right) and differences between methods are even more apparent. Thus, ViTs without Eureka-moment do not simply learn to ignore regions of the image. More datasets are described in A.9.

**Is this a vision or feature extraction problem?** To show that Eureka-moments transcend mere artifacts of vision data, we also show their occurrence in the context of simplistic algorithmic tasks, referred to as the **reasoning task**. Due to the simplicity of the task, we use a single-layer 4 head transformer model. Fig. 7a reveals that Eureka-moments appear even in this simplified task, which does not require any feature learning. Both HT and NormSoftmax reduce the training steps required for Eureka-moments to occur and increase the Eureka-ratio from 3/5 to 4/5 or 5/5, respectively.

**Is this a mere artefact of a bad choice of hyper-parameters?** We always use the default of 5 Warmup epochs to avoid training instabilities during early stages of training. We found that 20 Warmup epochs were most effective in mitigating the problem. However, sensitivity to the learning rate schedule (Tab. 2) is high. The average Eureka-epoch is very late (Tab. 1 left) and found more Warmup epochs lead to worse results on harder tasks (Tab. 1 right). The Eureka-ratio is sensitive to the learning rate schedule. We test 9 learning rate schedules for each method, (see Tab. 11). Lower temperatures are less sensitive to the learning rate schedule (see Tab. 2).

Weight Decay (WD) can facilitate grokking (Power et al., 2022) by forcing the network to learn general mechanisms (Power et al., 2022; Nanda et al., 2023). In our setting, we only found mild improvements for higher WD. However, more random seeds revealed, that higher WD rather reduces the Eureka-ratio (Tab. 1) and does not help in solving transformer's learning issue.

**Influence of model scale on Eureka-ratio**. We found no consistent influence of model scale on Eureka-ratio. More details are provided in A.5.

**Can the problem be fixed by rescaling of the gradient magnitude for $W_V$, $W_Q$ and $W_K$?** The observation that lower gradient imbalance leads to higher Eureka-ratio suggests, that simply rescaling of the gradients may solve the problem. We find that this also does not work consistently

Table 2: **Sensitivity to learning rate schedule**. Lower temperatures and in particular NormSoftmax drastically increase robustness to imperfect learning rate schedules. Eureka-ratio computed over seeds and learning rate schedules.

| Model | Eureka-ratio ↑ |
|---|---|
| ViT | 04/36 |
| ViT + Warmup 20 | 14/36 |
| $W_{QKV}$ grad scaling | 5/36 |
| NormSoftmax | **36/36** |
| ViT $\tau = \frac{1}{3}$ | 20/36 |
| ViT+HT $\frac{1}{3} \to \sqrt{d_k}$ | 25/36 |

(see Tab. 1) and is very sensitive to the learning rate (see Tab. 2). We attribute this to the differences in gradient magnitudes for indicators and targets and discuss it further in Sec.A.4.

**Do gradients vanish completely and can transformers recover?** Fig. 1b already suggests, that one potential solution to reliably get Eureka-moments is very long training. This observation is supported by Fig. 5, which indicates that gradients become small, but not 0. Indeed we observe that training for 3000 epochs results in an Eureka-ratio of 4/4 for all the learning rate schedules. In practice, this is of little help because the number of sub-tasks is unknown and Eureka-moments hard to predict.

**Does the NormSoftmax intervention also help on real datasets?** Jiang et al. (2022) reported improved performance and faster convergence on ImageNet and machine translation tasks using NormSoftmax. Both tasks likely contain some innate multi-step tasks, e.g. identifying a common discriminative feature and then discriminating between the difficult classes for ImageNet. Improvements may be due to easier multi-step learning with NormSoftmax. Deshpande & Narasimhan (2020) showed that attention learning slows down task 2 learning for RoBERTa. Indeed, training RoBERTa (Liu et al., 2019) with NormSoftmax leads to earlier Eureka-moments (see Fig 7b, more details in A.14). Thus, our analysis and results appear to be transferable to real datasets.

## 6 Limitations and Conclusion

**Limitations.** The ability to decompose tasks into sub-problems and learn to solve those sub-tasks is a common problem, but it is difficult to study on real datasets, since there are many confounders. As a result many works follow a trial and error approach. In contrast, we try to derive deeper understanding by studying this problem in isolation on synthetic data. This comes with the (implicit) assumption that our analysis transfers to real data, but we can provide only indications for that. We believe that both approaches provide different information that can be put together to make progress.

**Conclusion.** In this work, we identified that transformers have difficulties to decompose a task into sub-problems and learn to solve the intermediate sub-tasks. We observe that transformers can learn these tasks suddenly and unexpectedly but usually take a long time to do so. We called these Eureka-moments. We pin down the problem to the Softmax in the attention that leads to small gradients. We propose simple solutions that specifically target the Softmax and show that they improve the transformers' capabilities to learn sub-tasks and to learn them faster. We identify NormSoftmax as most robust and convenient method, leading to consistently better results.

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
