## A    APPENDIX

Here, we provide additional information that supports understanding and helps interpreting the main paper. We provide supplemental experimental results and more detailed analysis. We show the gradient norm for indicators and targets individually, which reveals that most of the already small gradients for $W_Q$ and $W_K$ is attributed to target features for models that do not learn the task and very little to indicator features. We provide the linear probe plots using also $Q$, $K$ and $V$ representations and the full version of Fig. 5. Next, we provide an ablation on model scale and the main results from the main paper with standard deviation and training speed up. We explain additional datasets and report results on them. We providing a more complete version of Fig. 5. Last, we provide details on training and experimental setups and an explanation for vanishing gradients in case of centralized attention maps.

### A.1    ATTENTION DISTRIBUTION OVER TIME.

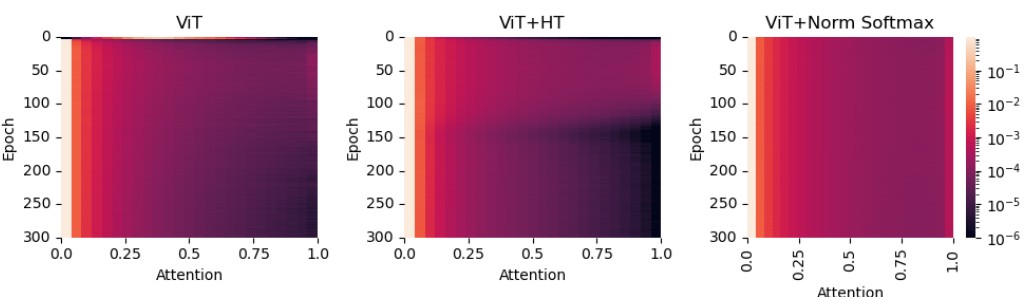

Figure 8: **Attention distribution as a heatmap.** Attention scores are sampled during evaluation after each epoch and binned to 25 bins. The color map is log scaled. For all 3 models, the vast majority of values falls into the first bin. ViT shows very few higher attention scores. ViT+HT and ViT+NormSoftmax lead to a significantly larger number of medium and high attention values. For ViT+HT this is limited to the first 100 epochs.

### A.2    GRADIENT NORM FOR INDICATORS AND TARGETS.

Our particular dataset design allows us to look at the gradients for indicators and targets separately. In particular, we make use of the fact, that indicators and targets are always at the exact same spacial location. More precisely, we use the partial derivatives as proxy for the gradients. We compute $\frac{\partial Z}{\partial Q}$, $\frac{\partial Z}{\partial K}$ and $\frac{\partial Z}{\partial V}$, where $Z$ is the output of the attention function. To analyze the gradient norm for targets and indicators independently we compute $\frac{\partial Z}{\partial Q}$, $\frac{\partial Z}{\partial K}$ and $\frac{\partial Z}{\partial V}$, where $Z$ is the output of the attention function. Since we compute the derivative wrt. the tokens, the spacial dimension remains. By averaging over the batch dimension and heads we can plot the partial derivative for each token. While it's not exactly the same, we will use the term gradient to refer to these partial derivatives in the following.

Since each token corresponds to a region in the image, we can visualize these results as an image. The results are shown in Fig. 3, Fig. 9 and Fig. 10. It can clearly be seen, that target regions (top-right and bottom-left) receive more gradients than indicator regions for $K$ and $V$. ViT, ViT+HT and NormSoftmax mitigate this problem, leading to significant gradient for indicator tokens. Indicator regions for $Q$ receive comparatively larger gradients, however, the gradients for $Q$ are much smaller.

Besides that, we compute the mean partial derivative for indicator and target regions of the image i.e. we average the partial derivatives for tokens corresponding to target regions or indicator regions. This allows us to plot the gradient norm for $Q$, $K$ and $V$ for only target and indicator tokens over the training. We show the results in Fig. 11. We make the following observations:

1. In general, the gradients are not evenly divided between target and indicator $K$, with usually smaller gradient for the indicator regions. Therefore Fig. 5 even underestimates the difference for the indicator regions, i.e. the regions relevant for an Eureka-moment.

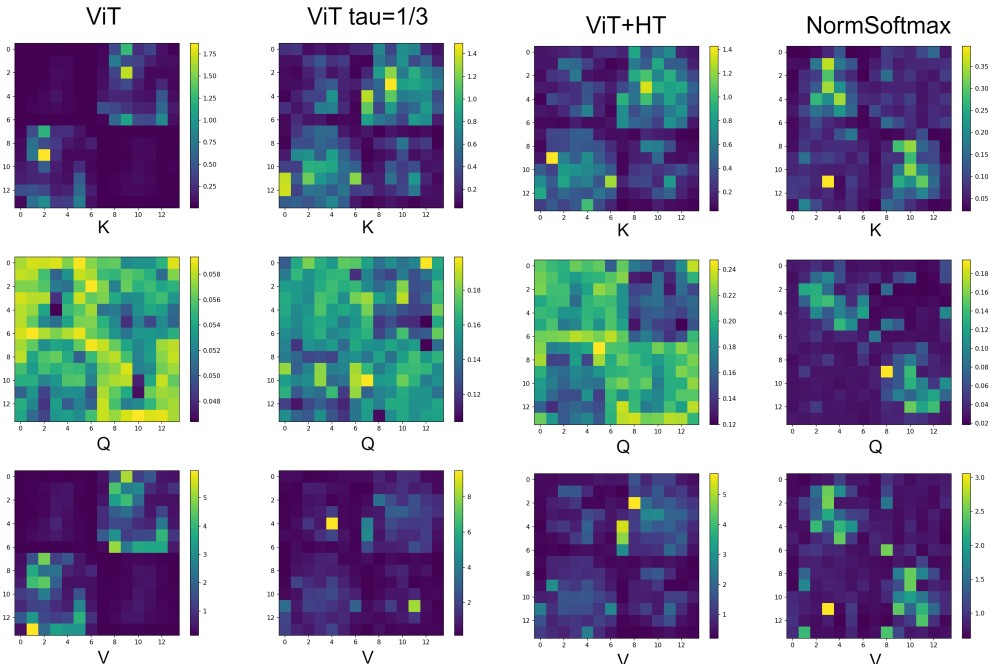

Figure 9: **Gradient norm for different image regions visualized for** $Q$, $K$, $V$**.** Larger gradients are visible for target regions, i.e, top-right and bottom-left. Indicator receive less gradients for $K$ and $V$. ViT $\tau = \frac{1}{3}$, ViT+HT and NormSoftmax mitigate this problem well. Indicator regions for $Q$ receive more gradient relative to the target regions, but overall the gradient for $Q$ is very small (see color bar). Plots created at epoch 50.

2. This difference can explain why more time is needed to reach an Eureka-moment for ViT in comparison to the other methods.

3. This difference between indicator $k$ and target $k$ can also explain, why the $W_{QKV}$ grad scaling does not solve the problem.

4. For ViT the difference between gradients of "$V$ and $K$" and "$V$ and $Q$" is, for most layers, generally much larger compared to the other approaches. This is particularly true before the Eureka-moment, where larger gradients for indicator $K$ and indicator $Q$ are crucial to get an early Eureka-moment. Most prominent is the difference between target $V$ and indicator $K$, showing a large mismatch. This explains, why the attention maps change so slowly and why simply increasing the learning rate does not solve the problem.

## A.3 GRADIENT NORM

Fig. 12 shows the L1 gradient norm for all layers and all methods. It can be seen, that throughout all layers and the entire training ViT has larger gradients for $W_V$ in comparison to $W_K$ and $W_Q$. For the other methods differences are much smaller and less consistent.

## A.4 WHY GRADIENT MAGNITUDE SCALING DOES NOT WORK.

Following the observation of Fig. 5, it stands to reason to simply scale the gradients for $W_V$, $W_Q$ and $W_K$ to the same value. To this end, we compute the gradient norm for $W_V$, $W_Q$ and $W_K$ for each layer and the overall mean. We scale the gradients for $W_V$, $W_Q$ and $W_K$, such that their norm is equal to the mean norm. This removes the imbalanced gradient issue and results in identical effective learning rate.

While this approach might work, it solves only part of the problem. Different features might receive differently large gradients. For instance the indicator features (here digits) receive little gradient,

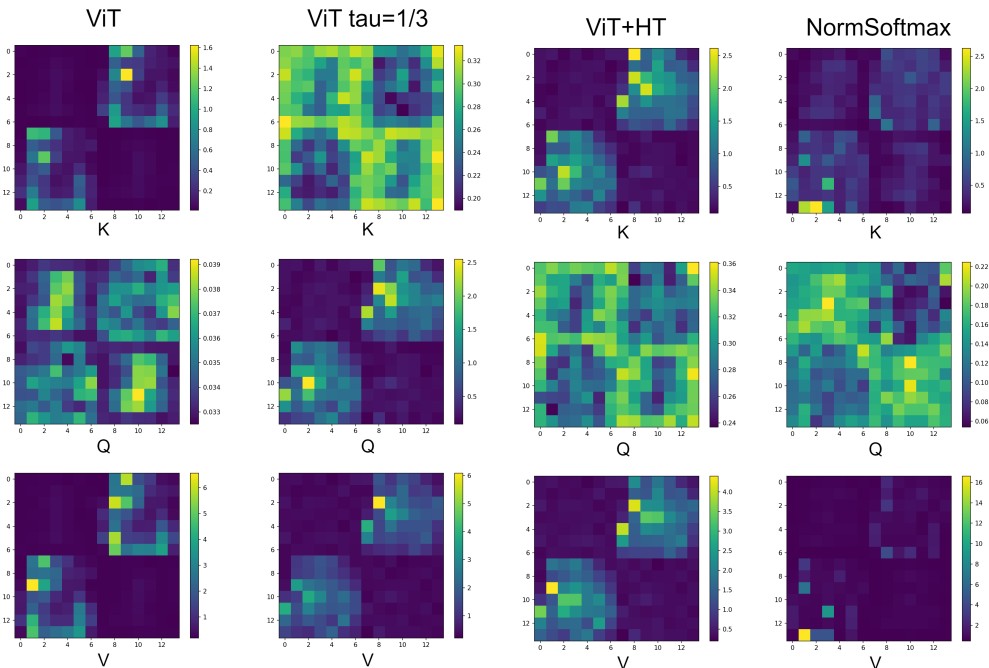

Figure 10: **Gradient norm for different image regions visualized for** $Q$, $K$, $V$. at epoch 13 (Eureka-moment of NormSoftmax).

while target (here fashion) receive large gradients, as can be seen in Fig. 9. Simply scaling up the gradients would not solve the imbalance between indicator and target gradients.

We can see in Tab. 1, that $W_{QKV}$ grad scaling helps on the main dataset, but is very sensitive to the learning rate (Tab.2). However, it completely fails on the harder task. The learning rate sensitivity and the failure on the harder task are most likely due to the gradient imbalanced discussed above.

### A.5    INFLUENCE OF MODEL SCALE ON EUREKA-RATIO.

The low Eureka-ratio could also be due to a too large or too small architecture. Also the number of heads might play an important role, since different features can be attended in different heads. More heads might increase the likelihood of one head specializing in indicators. Also the embedding dimension per head might just be too small or large for the task at hand. Maybe, even the hidden dimension of the MLP at the end of the attention block is the bottleneck. Many parameters of the transformer itself could explain why it fails to solve our tasks. We test these hypotheses in Tab. 3. While most changes lead to a lower Eureka-ratio, reducing the depth and increasing the number of heads leads to mild improvements. Combining both leads to an Eureka-ratio of 4/4, but, as can be seen in Tab. 5, this architecture does not generalize to other datasets.

### A.6    LINEAR PROBE RESULTS FOR $Q$, $K$ AND $V$ AND FOR TARGET CLASSIFICATION.

In the following we will show the linear probe results for $Q$, $K$, $V$ and $Z$ for all layers.

**Linear probe results for** $Z$ **for all layers.** Fig. 13 shows the same plot as in the main paper, but for all layers. In addition to the observations made for the main paper, we can see that layer 2, 3 and 6 for the ViT with Eureka-moment (Fig. 13b) represent significantly more information about the target locations than other layers in the CLS token. Indicating, that this information is extracted in these layers and written on the CLS token.

**Linear probes for** $Q$ **and** $K$ **and** $V$**.** Fig. 16 and Fig. 17 show the results when using the $Q$ and $K$ as input for the linear probes. Note, that $Q$ and $K$ are not updated by the residual connection of the attention block, therefore, no bars for "after residual" are plotted. The linear probe classification

Table 3: **Influence of model scale on Eureka-ratio**. Eureka-ratio is only partially influenced by the architecture. More shallow models and more heads both improve the results. The combination leads to 4/4 Eureka-ratio, but as can be seen in Tab. 5 this architecture fails at other tasks, while our solutions lead to improvements even on the "no position task".

| | | | | | Avg. over subset with Eureka-moment | |
| Heads | Emb. Dim. | Depth | MLP | Eureka-ratio ↑ | Accuracy ↑ | Avg. Eureka-epoch |
|---|---|---|---|---|---|---|
| 4 | 64 | 7 | 2 | 2/4 | $88.61 \pm 1.64$ | $232.50 \pm 35.50$ |
| 4 | 64 | 7 | 4 | 1/4 | $90.16 \pm 0.00$ | $162.00 \pm 0.00$ |
| 4 | 64 | 4 | 2 | 3/4 | $88.77 \pm 1.20$ | $216.33 \pm 46.64$ |
| 4 | 64 | 10 | 2 | 2/4 | $89.93 \pm 0.26$ | $221.00 \pm 18.00$ |
| 4 | 48 | 7 | 2 | 1/4 | $90.18 \pm 0.00$ | $137.00 \pm 0.00$ |
| 4 | 96 | 7 | 2 | 2/4 | $89.76 \pm 0.03$ | $140.50 \pm 70.5$ |
| 2 | 64 | 7 | 2 | 0/4 | - | - |
| 6 | 64 | 7 | 2 | 3/4 | $89.67 \pm 0.25$ | $139.67 \pm 55.16$ |
| 6 | 64 | 4 | 2 | **4/4** | $\mathbf{89.75 \pm 0.24}$ | $152.25 \pm 44.49$ |

accuracies for $Q$ and $K$ are very similar. Again, we can see that the ViT without Eureka-moment does not represent indicator information in the CLS token and target location can not be linearly separated from other information. Similarly, as for linear probe results with $Z$, layer 2, 3 and 6 for $Q$ and $K$ contain significantly more information about the indicators and target location for the ViT with Eureka-moment (compare Fig. 13b, Fig. 16b, Fig. 17b).

The linear probe results for $V$ look very similar to those for $Q$ and $K$ (see Fig. 18).

**Linear probes from $Z$ to targets.** Last, we show the linear probe results when predicting the targets from $Z$. As can be seen, from the entire representation, for both ViT with Eureka-moment and Vit without Eureka-moment, target classes can be predicted with high accuracy. Differences can be observed when using only the CLS token. Here, we observe that more target information is in the CLS token of the model without Eureka-moment (see Fig. 19).

### A.7   MAIN RESULTS WITH STANDARD DEVIATION, RESNET AND CONVERGENCE SPEED IMPROVEMENTS

Due to space and readability constraints we report in the main paper only the mean over all seeds. In Tab. 4 and Tab. 5 we show the same tables including the standard deviation.

Additionally, Tab. 4 and Tab. 5 also provide a comparison to a ResNet9.

Last, for Tab. 4 we report the improved convergence speed as a percentage of the number of training steps to reach 95% of ViT accuracy (averaged only over seeds with Eureka-moments), denoted as "% of steps". This value is computed only over the fraction of seeds, that actually lead to a higher accuracy than 95% of ViTs accuracy. In the last column, we also report this fraction. Note, that the Eureka-ratio is the maximum possible value for the "95%-ratio", i.e. for "ViT +Warmup 20" 8/10 seeds have an Eureka-moment. Out of these 8 only 5 reach an accuracy higher than 95% of the ViT accuracy.

### A.8   TRANSFORMERS LEARN THE PRIOR $p(z)$

Given a task like $p(y|x, z) \cdot p(z|x)$, i.e. the probability of class $y$ given evidence $x$ and the latent variable $z$, we argue, that transformers first learn a prior $p(z)$, ignoring the evidence. Sometimes they fail to unlearn this and pay attention to the evidence. In all previous experiments, the probability of target 1 or target 2 being the target to classify was 0.5. In a setting without 0.5 probability, the transformer should pick the target which is more frequently correct, in case it actually learns the prior $p(z)$. We test this by changing the probability of the top-right target to be the target location to 0.65. As can be seen in Fig. 14, the transformer initially learns the shortcut of always picking the more likely target.

Table 4: **Main dataset – Comparison of proposed solutions and baselines.** This is a complete version of Tab. 1 including standard deviation, and speed improvements. For the **main dataset**, as described in Fig. 1a. **ER**: Eureka-ratio, **Acc.**: Accuracy, **Avg. EE.**: average Eureka-epoch. **% of steps** indicates the % of steps needed to reach 95% of ViTs accuracy. **95%-ratio** indicates the ratio of models that actually reached 95% of ViTs accuracy.

| | | | Main Dataset | | | |
|---|---|---|---|---|---|---|
| | | | Avg. over EMs | | Avg. over ViT 95% Acc. | |
| Model | $\tau$ | ER $\uparrow$ | Acc. $\uparrow$ | Avg. EE. | % of steps | 95%-ratio |
| ResNet | | 10/10 | $99.40 \pm 0.10$ | $3.00 \pm 00.00$ | - | - |
| ViT | $\frac{1}{0.025}$ | 3/10 | $89.40 \pm 0.08$ | $174.67 \pm 37.82$ | 84.26 | 3/10 |
| ViT | $\frac{1}{0.075}$ | 6/10 | $90.13 \pm 0.40$ | $181.34 \pm 24.94$ | 86.79 | 2/10 |
| ViT + WD 0.5 | $\sqrt{d_k}$ | 5/10 | $90.09 \pm 0.27$ | $177.80 \pm 52.30$ | 84.70 | 5/10 |
| ViT | $\sqrt{d_k}$ | 7/10 | $89.48 \pm 1.10$ | $207.43 \pm 46.65$ | 100.00 | 4/10 |
| ViT + Warmup 20 | $\sqrt{d_k}$ | 8/10 | $87.65 \pm 6.48$ | $205.87 \pm 57.05$ | 91.87 | 5/10 |
| $W_{QKV}$ grad scaling | $\sqrt{d_k}$ | **10/10** | $87.96 \pm 2.45$ | $119.4 \pm 62.70$ | 73.79 | 5/10 |
| NormSoftmax | $\sqrt{d_k}$ | **10/10** | $89.56 \pm 0.65$ | $28.20 \pm 34.85$ | 19.87 | 10/10 |
| NormSoftmax | $\frac{1}{3}$ | **10/10** | $89.18 \pm 0.36$ | $23.50 \pm 08.15$ | 19.25 | 10/10 |
| ViT | $\frac{1}{3}$ | **10/10** | $89.35 \pm 0.28$ | $66.60 \pm 58.55$ | 36.71 | 10/10 |
| ViT+HT | $\frac{1}{3} \to \sqrt{d_k}$ | **10/10** | **89.81** $\pm$ **0.29** | $74.00 \pm 61.29$ | 39.88 | 10/10 |
| NormSoftmax + HT | $\frac{1}{3} \to \sqrt{d_k}$ | **10/10** | $89.83 \pm 0.41$ | $17.50 \pm 04.84$ | 16.41 | 10/10 |

Table 5: **No Position Task – Comparison of proposed solutions and baselines.** For the **No position task**, as described in Fig. 6b. $\tau$ not optimized for this task. **ER**: Eureka-ratio, **Acc.**: Accuracy, **Avg. EE.**: average Eureka-epoch.

| | | | No Position Task | |
|---|---|---|---|---|
| | | | Avg. over EMs | |
| Model | $\tau$ | ER $\uparrow$ | Acc. $\uparrow$ | Avg. EE. |
| ResNet | | 4/4 | $91.27 \pm 0.30$ | $4.25 \pm 00.43$ |
| ViT | $\sqrt{d_k}$ | 0/4 | - | - |
| ViT + Warmup 20 | $\sqrt{d_k}$ | 1/4 | $89.55 \pm 0.00$ | $117 \pm 00.00$ |
| $W_{QKV}$ grad scaling | $\sqrt{d_k}$ | 0/4 | - | - |
| NormSoftmax | $\sqrt{d_k}$ | **3/4** | **88.98** $\pm$ **0.55** | **228.67** $\pm$ 08.22 |
| NormSoftmax | $\frac{1}{3}$ | 1/4 | $89.77 \pm 0.00$ | $20.00 \pm 00.00$ |
| ViT | $\frac{1}{3}$ | 1/4 | $89.68 \pm 0.00$ | $191.00 \pm 00.00$ |
| ViT+HT | $\frac{1}{3} \to \sqrt{d_k}$ | 1/4 | $88.36 \pm 0.00$ | $242.00 \pm 00.00$ |
| NormSoftmax + HT | $\frac{1}{3} \to \sqrt{d_k}$ | 1/4 | $90.63 \pm 0.00$ | $19.00 \pm 00.00$ |
| ViT 6 heads, depth 4 | | 0/0 | - | - |

## A.9 DESCRIPTION OF AND RESULTS ON MORE DATASETS

In the following we report results on 5 more datasets. The datasets are depicted in Fig. 15.

**Cifar task 1.** A schematic for this task is shown in Fig. 15a. The "Cifar task 1" dataset uses Cifar-10 Krizhevsky et al. (2009) images of classes "automobile" and "bird" as indicators. Targets are sampled from fashion MNIST and MNIST. All 4 images are randomly placed on a 4x4 canvas and we apply random colors (red or blue) to the MNIST and fashion MNIST samples. Task 1 is to compare the Cifar-10 classes. If they come from the same class, task 2 is to classify the fashion MNIST sample. If not, the tasks is to classify the MNIST digit. Results are reported in Tab. 6. normal ViT fails in 1/4 cases and Eureka-epoch is usually late. Note, that this task may seem difficult, but differences in color distribution of "bird" and "automobile" simplify the task.

**Top if above.** The task description is summarized in Fig. 15b. For data creation we sample 2 images from fashion MNIST and place one in the top row of a 4x4 canvas and the other in the bottom row.

Table 6: **Results "Cifar task 1" dataset**. $\tau$ not optimized for this task. **ER**: Eureka-ratio, **Acc.**: Accuracy, **Avg. EE.**: average Eureka-epoch.

| | | | Cifar task 1 | |
| --- | --- | --- | --- | --- |
| | | | Avg. over EMs | |
| Model | $\tau$ | ER ↑ | Acc. ↑ | Avg. EE. |
| ViT | $\sqrt{d_k}$ | 3/4 | $83.43 \pm 1.83$ | $187.67 \pm 42.46$ |
| ViT | $\frac{1}{3}$ | **4/4** | $\mathbf{86.86 \pm 0.75}$ | $\mathbf{76.50 \pm 08.90}$ |
| NormSoftmax | $\sqrt{d_k}$ | **4/4** | $82.89 \pm 0.31$ | $100.0 \pm 27.89$ |

The column is selected randomly for both. Task 1 is to check whether the 2 samples are in the same column. If they are task 2 is to classify the top image. If not, the image in the bottom row must be classified. This task is relatively simple, as it removes additional indicators. Instead, relative location of the images is the relevant information to solve task 1. This task is very simple and leads to a low Eureka-epoch for all methods (see 15b).

Table 7: **Results "Top if above" dataset**. $\tau$ not optimized for this task. **ER**: Eureka-ratio, **Acc.**: Accuracy, **Avg. EE.**: average Eureka-epoch.

| | | | Top if above | |
| --- | --- | --- | --- | --- |
| | | | Avg. over EMs | |
| Model | $\tau$ | ER ↑ | Acc. ↑ | Avg. EE. |
| ViT | $\sqrt{d_k}$ | **4/4** | $\mathbf{91.47 \pm 0.13}$ | $13.75 \pm 2.19$ |
| ViT | $\frac{1}{3}$ | **4/4** | $90.38 \pm 0.13$ | $9.5 \pm 1.25$ |
| NormSoftmax | $\sqrt{d_k}$ | **4/4** | $90.84 \pm 0.27$ | $\mathbf{9.25 \pm 1.09}$ |

**Same color decision task.** The task is explained in Fig. 15b. For data creation we sample only MNIST digits and apply random colors (red or blue) to all digits. If color of the indicators is identical, the top right must be classified and bottom left if not. As can be seen in Tab. 8, this task is again very easy. Color seems to be easily accessible for ViT and ViT has little trouble to compare the indicator colors.

Table 8: **Results "Same color decision task" dataset**. $\tau$ not optimized for this task. **ER**: Eureka-ratio, **Acc.**: Accuracy, **Avg. EE.**: average Eureka-epoch.

| | | | Same color decision task | |
| --- | --- | --- | --- | --- |
| | | | Avg. over EMs | |
| Model | $\tau$ | ER ↑ | Acc. ↑ | Avg. EE. |
| ViT | $\sqrt{d_k}$ | **4/4** | $98.35 \pm 0.58$ | $91.5 \pm 60.04$ |
| ViT | $\frac{1}{3}$ | **4/4** | $\mathbf{98.95 \pm 0.10}$ | $8.25 \pm 1.48$ |
| NormSoftmax | $\sqrt{d_k}$ | **4/4** | $98.93 \pm 0.03$ | $\mathbf{7.75 \pm 0.43}$ |

**Color or fashion classification.** This task is shown in Fig. 15d. For the creation of the dataset we define 10 random colors, i.e. (brown, blue, yellow, orange, red, green, purple, gray, pink, turquoise) and apply a random color to each target and each indicator sample. For targets we use fashion MNIST samples and indicators are MNIST classes 1 and 2. Task 1 is to compare digits. If they are the same class, the top right fashion sample must be classified. If not, the color of the top right sample must be classified.

**Digit grouping.** Finally, we make the indicator task more difficult. We follow the same setting as for the "main dataset", as described in Fig 1a. However, indicators are not sampled from digits 1 and 2,

Table 9: **Results "Color or fashion class" dataset**. $\tau$ not optimized for this task. **ER**: Eureka-ratio, **Acc.**: Accuracy, **Avg. EE.**: average Eureka-epoch.

| | | | Color or fashion class | |
| --- | --- | --- | --- | --- |
| | | | Avg. over EMs | |
| Model | $\tau$ | ER $\uparrow$ | Acc. $\uparrow$ | Avg. EE. |
| ViT | $\sqrt{d_k}$ | 4/4 | $92.85 \pm 0.25$ | $12.00 \pm 3.32$ |
| ViT | $\frac{1}{3}$ | 4/4 | $92.53 \pm 0.40$ | $20.75 \pm 2.49$ |
| NormSoftmax | $\sqrt{d_k}$ | 4/4 | $\mathbf{92.75 \pm 0.71}$ | $\mathbf{11.25 \pm 0.83}$ |

but from 1, 2, 3 and 4. Task 1 is to find out whether both indicators are smaller or both indicators are larger or equal to 3. I.e. we build indicator sets $[1, 2]$ and $[3, 4]$ if both indicators are from the same group the top-right image should be classified. As can be seen in Tab. 10, increasing the difficulty of the indicator task quickly makes the dataset too hard. Further optimization of hyper-parameters and architecture are likely to solve the tasks.

Table 10: **Results "Digit grouping" dataset**. $\tau$ not optimized for this task. **ER**: Eureka-ratio, **Acc.**: Accuracy, **Avg. EE.**: average Eureka-epoch.

| | | | Digit grouping | |
| --- | --- | --- | --- | --- |
| | | | Avg. over EMs | |
| Model | $\tau$ | ER $\uparrow$ | Acc. $\uparrow$ | Avg. EE. |
| ViT | $\sqrt{d_k}$ | 0/4 | - | - |
| ViT | $\frac{1}{3}$ | 0/4 | - | - |
| NormSoftmax | $\sqrt{d_k}$ | 0/4 | - | - |

## A.10 EXPERIMENTAL SETUP — VISION TASK

We mostly follow the DeiT Touvron et al. (2021) training recipe without distillation. For optimization we use AdamW Loshchilov & Hutter (2019) with default values, i.e. $\beta_1 = 0.9$ and $\beta_2 = 0.999$ and $\epsilon = 10^{-8}$. Unless otherwise stated we Warmup the learning rate for 5 epochs from $10^{-6}$ to the maximum learning rate, use a weight decay of 0.05 and train for 300 epochs. We train all models with a batch size of 512, which fits on a single V100, for all the architectures that we considered. Since position and color can be important cues in our datasets we train without data augmentation. We only sample color-noise from a standard normal Gaussian with standard deviation 0.05 for each color channel independently. Color-noise is sampled for each sub-sample (i.e. each indicator and each target) independently and added to the RGB value.

**Learning rate schedules for all models.** To compare the tested methods and models fairly, we run a search over 9 learning rate schedules with 4 random seeds, each. We anneal the learning rate from a maximum to a minimum using a cosine scheduler. We also use Warmup, as described in the section on training details. The different schedules can be seen in Tab. 11. We pick the schedule that leads to highest Eureka-ratio for each model. In case of a tie we pick the schedule with higher accuracy.

**Info on taus:** In initial experiments we tested for ViT 5 $\tau$, $\tau = \frac{2}{3}$, $\tau = \frac{1}{2}$, $\tau = \frac{1}{3}$, $\tau = \frac{1}{4}$ and $\tau = \frac{1}{5}$. We found $\tau = \frac{1}{3}$, $\tau = \frac{1}{4}$ to work well and did not further optimize them for the different methods. For HT we set the goal temperature to the default $\sqrt{d_k}$ and tried also $\frac{1}{2*\sqrt{d_k}}$. Further optimizing these parameters for each model and dataset will most likely lead to improvements, but would add very little to a deeper understanding of the Eureka-moments.

Table 11: **Learning rate schedules.** We use cosine annealing from "max learning rate" to "min learning rate".

| max learning rate | min learning rate |
|---|---|
| $10^{-3}$ | $10^{-5}$ |
| $10^{-3}$ | $5 * 10^{-6}$ |
| $10^{-4}$ | $10^{-5}$ |
| $5 * 10^{-4}$ | $5 * 10^{-6}$ |
| $5 * 10^{-4}$ | $10^{-6}$ |
| $10^{-4}$ | $10^{-6}$ |
| $5 * 10^{-5}$ | $10^{-6}$ |
| $10^{-5}$ | $10^{-6}$ |
| $10^{-5}$ | $10^{-7}$ |

## A.11 IMPLEMENTATION DETAILS NORMSOFTMAX

In practice, $\sigma(\cdot)$ can be defined by arbitrary functions. As highlighted in the background section, the standard deviation is a theoretically motivated choice. Alternatives are discussed by Jiang et al. (2022). In this work, we find the variance to work better for ViT and RoBERTa, while we stick to the standard deviation for the reasoning task.

## A.12 EXPERIMENTAL SETUP – REASONING TASK

The input of the model is of the form "a b c d =", where, where $a, b, c, d \in \{0, 1, \ldots, n\}$. In our experiments, we set $n = 11$. We train the transformer on $30\%$ of the entire set of possible inputs (i.e., $11^4 = 14\,641$ input combinations), that is with a batch size of 4392. The rest is used as test set. We train for $10\,000$ epochs over five random seeds. We use token embeddings of size of $d = 2^{\lceil \log_2 n \rceil} = 16$, four attention heads of dimension of $d/4 = 4$, $4d = 64$ hidden units in the MLP, and learned positional embeddings.

We trained with full batch gradient descent using AdamW (Loshchilov & Hutter, 2019) with a cross-entropy loss. We optimized learning rates via grid search over $[10^{-4}, 10^{-2}]$ on seed 0. Following Nanda et al. (2023) we use a weight decay of 1.

## A.13 SLINGSHOT EFFECTS ON NON-REASONING TASK

We observed that NormSoftmax caused slingshot effects (Thilak et al., 2022) during the convergence phase of some of the training runs but believe this may be due to the interaction of gradients at different scales with adaptive optimizers (Nanda et al., 2023). Since slingshot effects only occur after Eureka-moments, they cannot be the cause for their occurrence. We did not further investigate this observation.

## A.14 EXPERIMENTAL SETUP ROBERTA

The RoBERTa experiments are based on the Code provided by (Deshpande & Narasimhan, 2020). We follow the data acquisition and preparation strategy of Shoeybi et al. (2019). Thus, we train on the latest Wikipedia dump (downloaded on 08.02.2023). We train a 12 layer RoBERTa model with 12 heads. We use a batch size of 84 and a learning rate of 5e-5.

### A.15 VANISHING GRADIENT IN THE SOFTMAX.

**Softmax attention can cause vanishing gradients for $W_q$ and $W_K$.** To see that softmax attention can result in vanishing gradients it helps to take a look at the gradients of the attention function. Let

$$A(W_q, W_k, W_v, X) = Z \tag{3}$$

$$A(W_q, W_k, W_v, X) = S\Big( D\big( Q(W_q, X), K(W_k, X)\big)\Big)(W_v, X), \tag{4}$$

$$S(D) = \text{softmax}(D), \tag{5}$$

$$D(Q, K) = \frac{QK^T}{\tau}, \tag{6}$$

$$Q(W_q, X) = W_q X, \tag{7}$$

$$K(W_k, X) = W_k X, \tag{8}$$

$$V(W_v, X) = W_v X. \tag{9}$$

be the attention function, where $W_k$, $W_q$ and $W_v$ are weight matrices, $X$ is the input.

Using the chain rule we get

$$\frac{\partial A}{\partial W_q} = \frac{\partial A}{\partial D}\frac{\partial D}{\partial Q}\frac{\partial Q}{\partial W_q} \tag{10}$$

$$\frac{\partial A}{\partial W_k} = \frac{\partial A}{\partial D}\frac{\partial D}{\partial K}\frac{\partial K}{\partial W_k}. \tag{11}$$

Since $\frac{\partial D}{\partial Q}, \frac{\partial Q}{\partial W_q}, \frac{\partial D}{\partial K}, \frac{\partial K}{\partial W_k}$ are constants we only need to look more closely into $\frac{\partial A}{\partial D}$.

$\frac{\partial A}{\partial D}$ is given by $\frac{\partial A}{\partial D} = \frac{\partial S}{\partial D}V$, where $S(D)$ takes the values $S = (s_1, \ldots, s_n)$. Therefore, to analyze how the gradients $\frac{\partial A}{\partial W_q}$ and $\frac{\partial A}{\partial W_k}$ behave, we need to analyze the $\frac{\partial S}{\partial D}$, i.e. the Jacobian of the Softmax $S(D)$. It is given by

$$\frac{\partial S}{\partial D} = \begin{pmatrix} s_1(1-s_1) & -s_1 s_2 & \ldots & -s_1 s_n \\ -s_2 s_1 & s_2(1-s_2) & \ldots & -s_2 s_n \\ \vdots & \vdots & \ddots & \vdots \\ -s_n s_1 & -s_n s_2 & \ldots & s_n(1-s_n) \end{pmatrix}. \tag{12}$$

It can be easily seen, that almost all entries in the Jacobian are close to 0 whenever a single $s_i$ is close to 1 and all others are almost 0.

### A.16 GRADIENT PLOTS FOR VIT WITH AND WITHOUT EUREKA-MOMENT

### A.17 EUREKA-MOMENTS ON REALISTIC, LARGE SCALE HIGH-RESOLUTION IMAGES

### A.18 ROBERTA TRAINING CURVE WITH VALIDATION LOSS

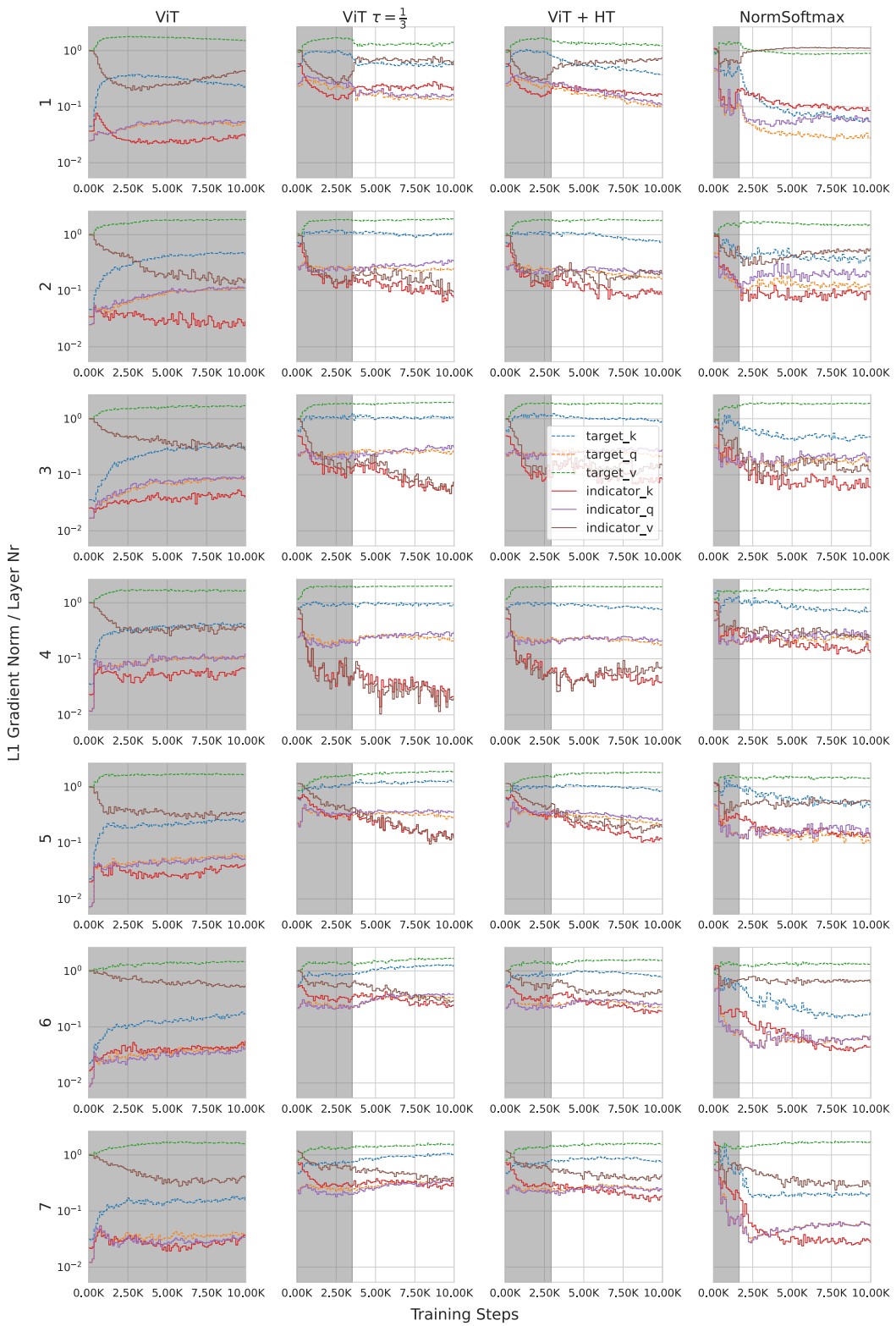

Figure 11: **L1 gradient norm separately for indicator and target tokens.** Indicator regions/features receive much less gradient than target regions. Gray region indicates steps before Eureka-moment.

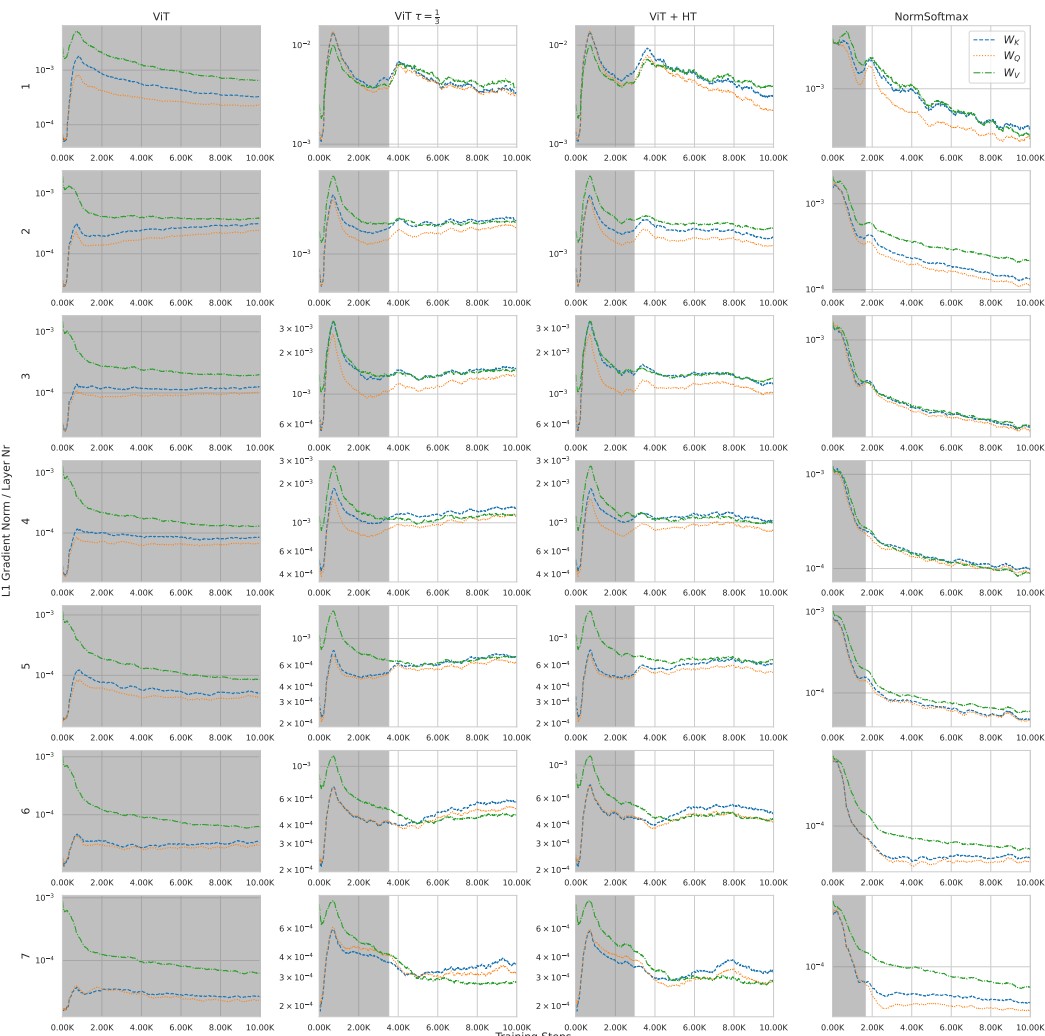

Figure 12: **L1 gradient norm plot for all layers, all models and entire training.** This is the complete version of Fig. 5. For ViT it can be seen that for all layers the gradient for $W_V$ is significantly larger than for $W_Q$ and $W_K$. For ViT+HT, ViT with $\tau = \frac{1}{3}$ and NormSoftmax the gradient norm is very similar for the weight matrices (note that the y-axis is not shared). NormSoftmax achives this also for deeper layers. Often after Eureka-moment $W_V$ starts to get larger attention than $W_K$ and $W_Q$. We conjecture that this is because the attention is already optimized, while task 2 can still improve by modifying the feature representation. Gray region indicates steps before Eureka-moment.

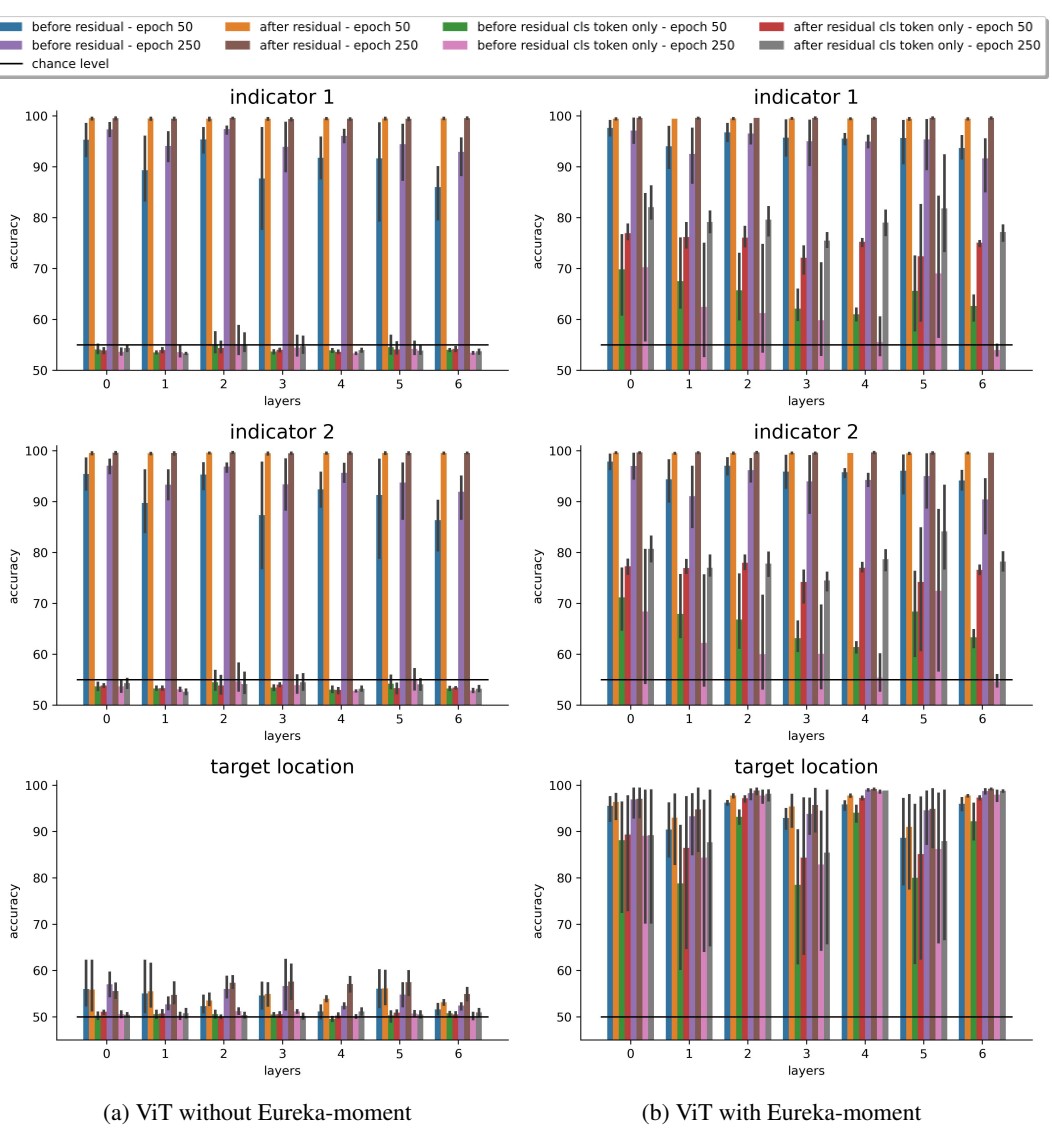

(a) ViT without Eureka-moment        (b) ViT with Eureka-moment

Figure 13: **Linear probe results for with $Z$ as input (all layers).** This the complete version of Fig. 2. Additionally, we can observe in (b) that layers 2, 3 and 6 contain more target location information than other layers, indicating, that this information can be processed in these layers.

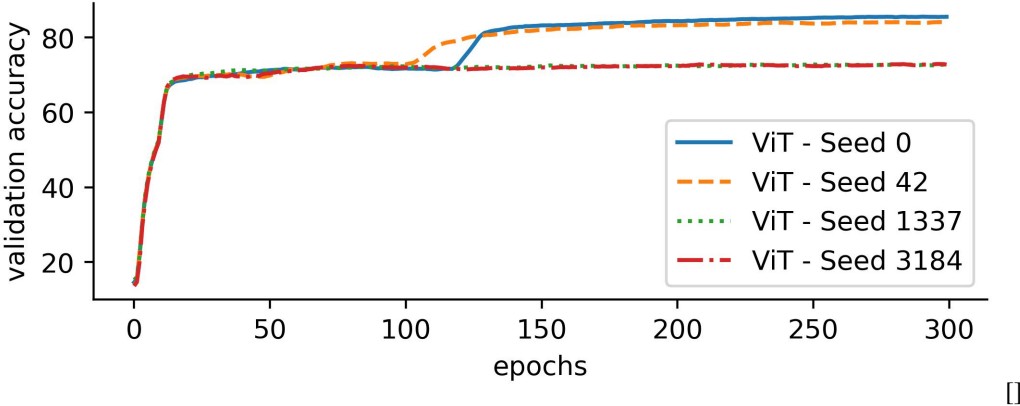

[]

Figure 14: **Validation accuracy curves for "main dataset" with changed target probabilities.** ViT learns that one target is more likely than the other and learn to always pick this target, as can be seen by the higher plateau accuracy.

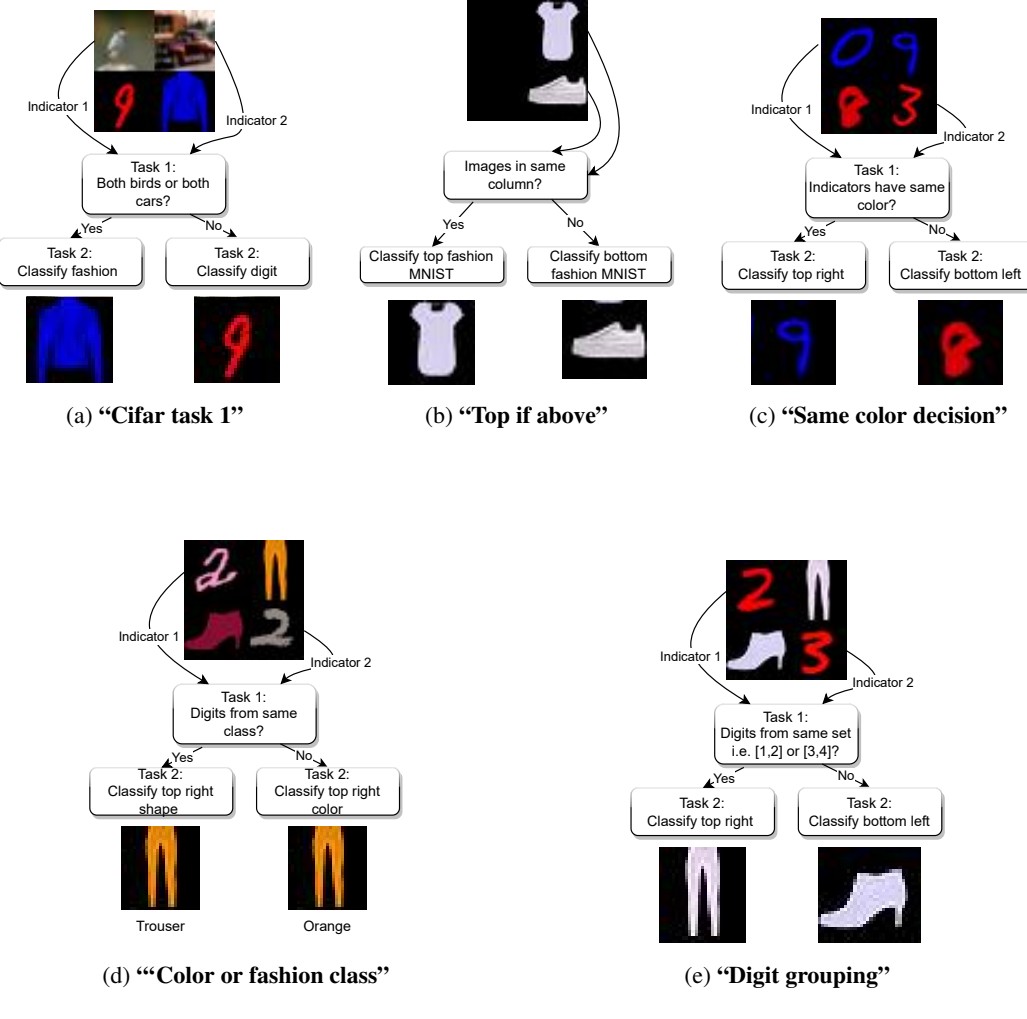

Figure 15: **Schematics for additional datasets.**

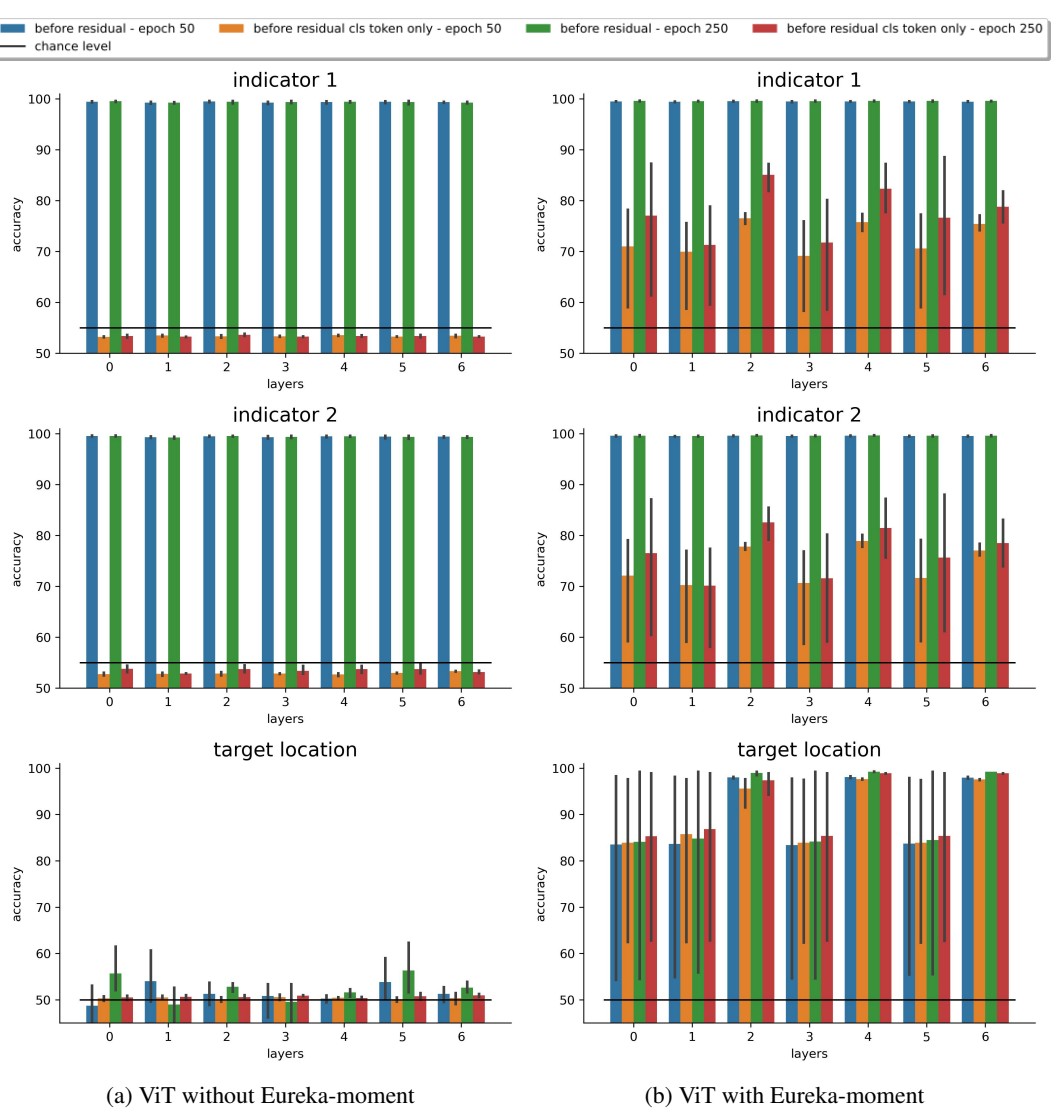

(a) ViT without Eureka-moment

(b) ViT with Eureka-moment

Figure 16: **Linear probe for Q.**

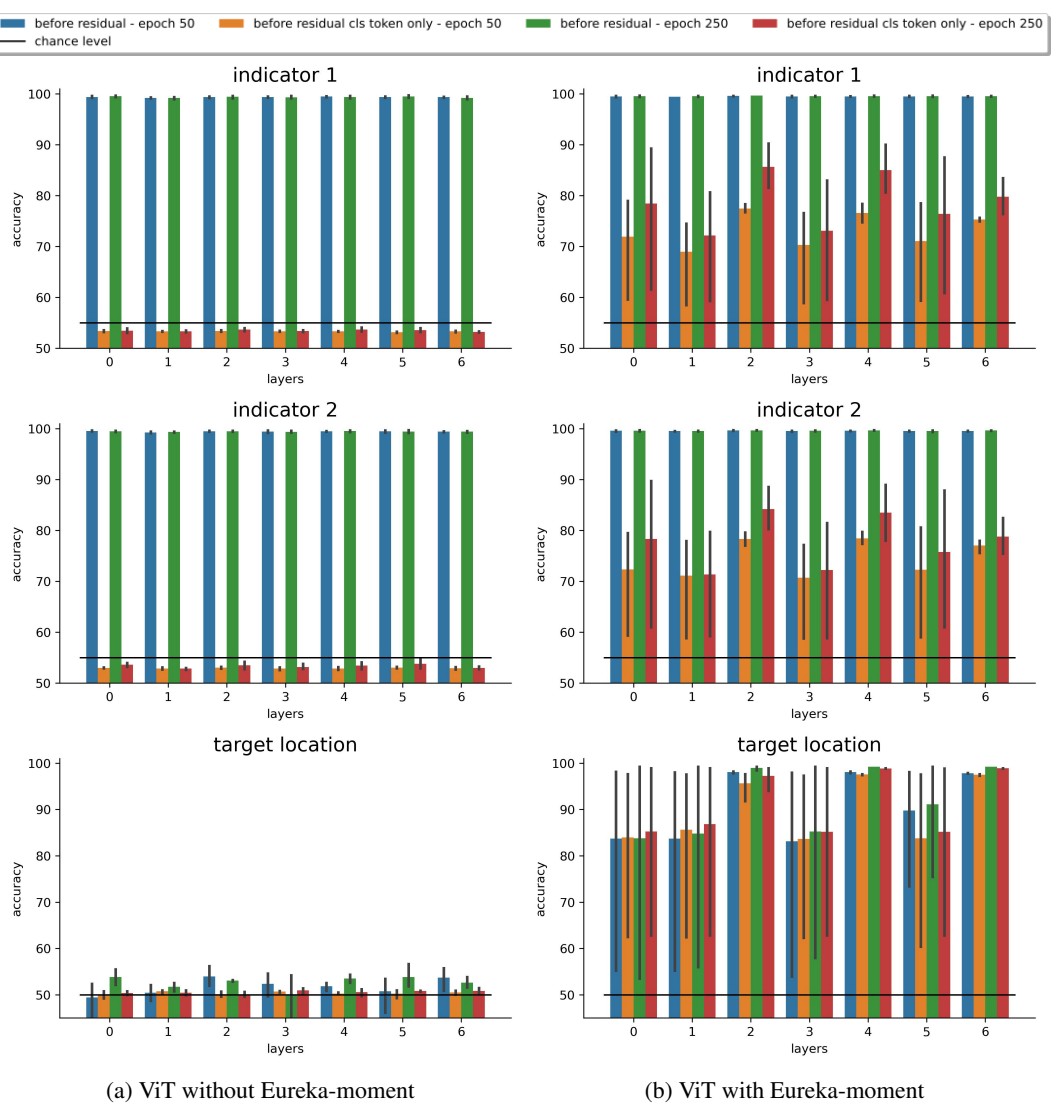

(a) ViT without Eureka-moment

(b) ViT with Eureka-moment

Figure 17: **Linear probe for K.**

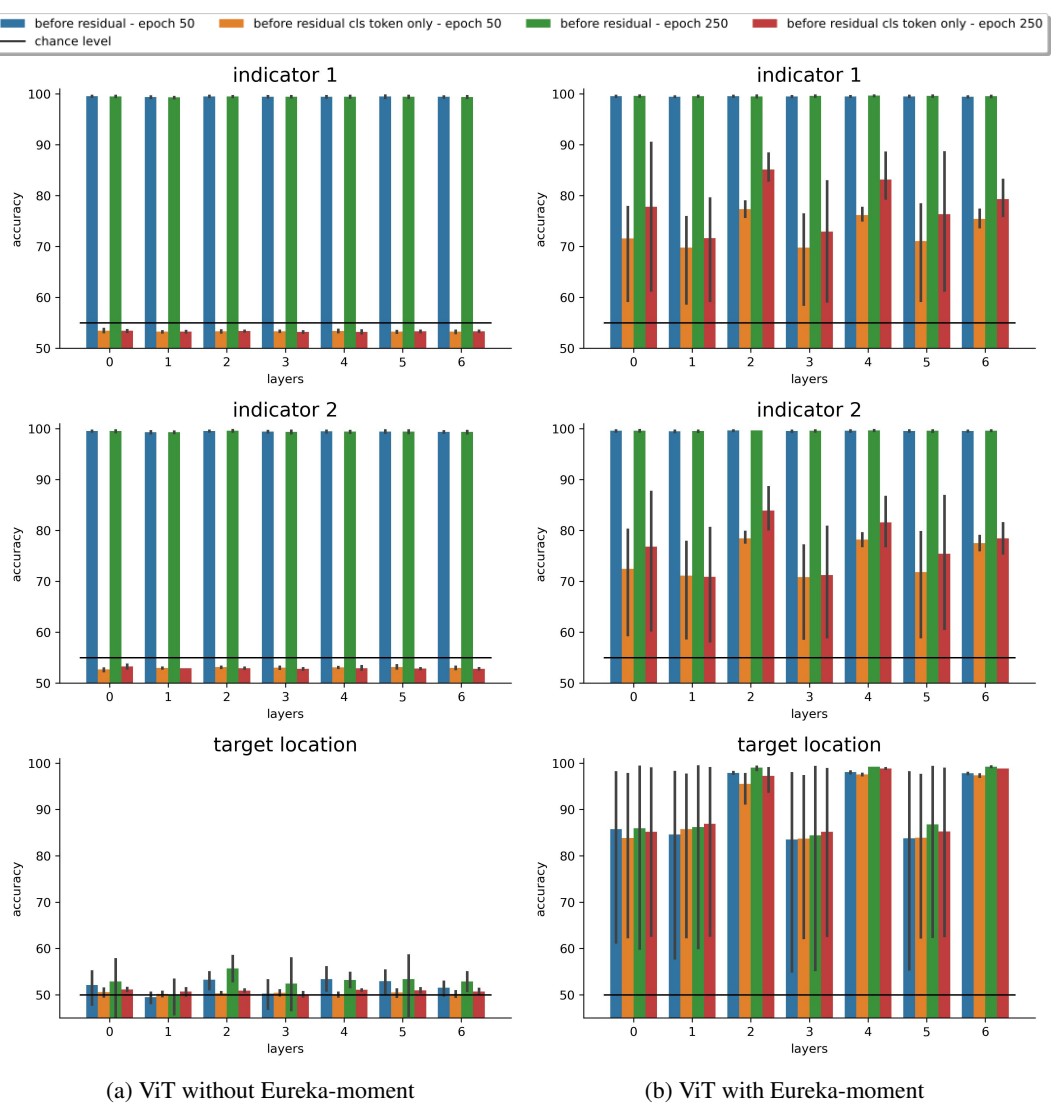

(a) ViT without Eureka-moment  (b) ViT with Eureka-moment

Figure 18: **Linear probe for V.**

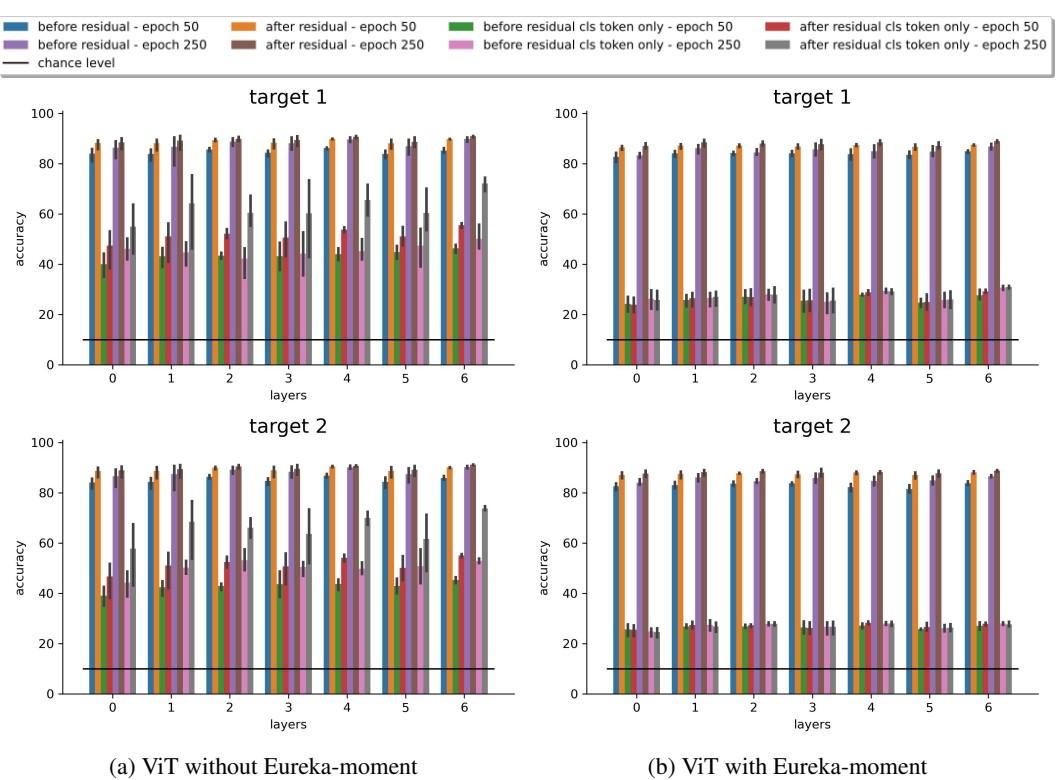

(a) ViT without Eureka-moment

(b) ViT with Eureka-moment

Figure 19: **Linear probe for Z with target classification.**

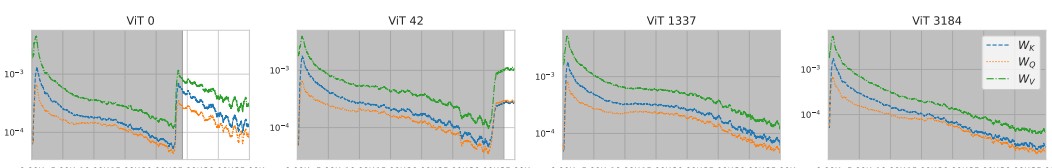

Figure 20: Gradients for $W_K$, $W_Q$ and $W_V$ for ViTs with different seeds. Gray box indicates the region before the Eureka-moment. ViTs all have very late Eureka-moments. We do not observe differences between training runs with and training runs without Eureka-moment. Number above the plots correspond to the random seeds.

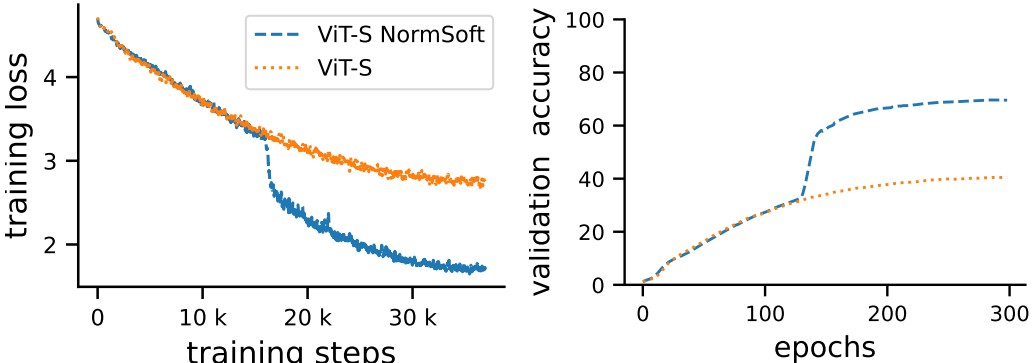

(a) Training loss on ImageNet-100 dog decision for ViT-S trained with NormSoftmax and vanilla ViT-s

(b) Validation accuracy on ImageNet-100 dog decision for ViT-S trained with NormSoftmax and vanilla ViT-s

Figure 21: Results on ImageNet-100 based task. Also for a larger ViT, realistic data, high-resolution images we observe Eureka-moments. In particular, NormSoftmax leads to a Eureka-moment, while the vanilla ViT fails to learn task 1. Task description: If both indicators show the image of an identical dog the top-right image is the target and bottom left otherwise. Dog samples taken from 2 dog classes. Probability of top-right being the target is set to 0.5.

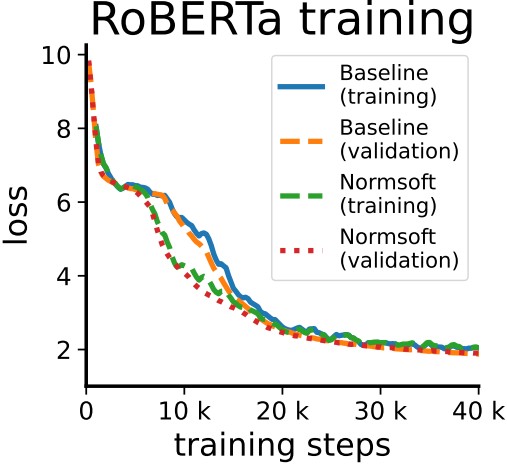

Figure 22: Complete version of Fig. 7b, including the validation loss curves.