# OpenReview forum: "Eureka-Moments in Transformers: Multi-Step Tasks Reveal Softmax Induced Optimization Problems"
_ICLR.cc/2024/Conference — Submitted to ICLR 2024_

### Official Review · Reviewer_Lp2g · 2023-10-31

**Soundness:** 2 fair
**Presentation:** 2 fair
**Contribution:** 2 fair
**Rating:** 5
**Confidence:** 3

**Summary:**

This study explores the challenges transformers face when confronted with multi-step decision tasks, in contrast to CNNs which show no difficulties with the same tasks. It was discovered that transformers can suddenly improve, or experience 'Eureka-moments', after their training and validation loss have been stagnant for hundreds of epochs. These moments differ from the known Grokking phenomena, as both validation and training loss plateau before suddenly improving during a Eureka-moment. The underlying issue was traced back to the Softmax function in the self-attention block of transformers.

**Strengths:**

- The paper is well-written and easy to follow.
- The paper has a interesting discovery that transformers can suddenly improve, or just can not converge.
- The paper present a fine-grained analysis on the problems that might be asked.

**Weaknesses:**

While the paper does not propose a new solution to the identified issue, the contribution lies predominantly in the realm of discovery, task design, and experimentation.

- In terms of discovery and task design, the paper points out an intriguing issue with the softmax function (leading to slow and unstable convergence), an issue that has been identified and addressed by the previously proposed NormSoftmax. Moreover, the task designed within this study does not align with practical application needs and seems to lack significant utility, serving more as a platform for theoretical experimentation. It might be more beneficial to design an experiment around a multi-step task that is more natural and more applicable to visual applications.

- The paper identifies a problem with traditional Transformers and suggests that NormSoftmax could mitigate this issue. Viewed from the perspective that the problem has already been solved, the paper's contribution lies in its detailed experimental analysis and the answers provided to some questions. However, most conclusions and answers are drawn from a speculative and "might be" perspective, lacking sufficient theoretical support or more reliable evidence. It is recommended to focus on the phenomena that provide the most insight and to provide stricter proof to serve as the core contribution of the paper.

- The tasks designed in the paper are based on simpler datasets like MNIST and FashionMNIST. There is a lack of experimentation with more diverse and realistic high-resolution image datasets. For example, constructing new tasks using datasets like ImageNet and CelebA could increase the credibility and universality of the research results. We expect that the conclusions drawn still be as applicable to more complex real-world data.

**Questions:**

See weekness.

---

> ### Author Response · Authors · 2023-11-20
> **Answer to Lp2g**
>
> We appreciate your careful review and helpful comments. We address your concerns and questions below and in the general comment to all reviewers.
>
> One of the weaknesses mentioned by Lp2g is the weak alignment with practical application needs and significant utility of our work. Indeed, this work does not aim at contributing a method or dataset for some application. Instead, this work aims at understanding existing methods, highlighting a shortcoming, identifying a previously unknown failure mode and providing an experimental setup that can be used to study this phenomenon. We argue that improvements in the understanding of learning behavior of models may also enhance application performance further down the line.
>
> We suspect that for many real world applications Eureka-Moments happen, but can not be easily observed. We suspect that tasks exist where the Eureka-Moment does not happen which results in severe underperformance.
> Knowing about Eureka-Moments will enable the community to understand training failures better and will guide future experiments to avoid such.
>
> We would also like to highlight that we do not claim that NormSoftmax is a solution to the problem. We use it as an intervention to show that the problem is caused by the Softmax Attention. Note that this is the gold standard technique to understand cause-effect relationships commonly used also in, e.g., medical drug testing.
>
> Regarding the lack of experiments on more diverse and realistic high-resolution image datasets, we refer to the general comment to all reviewers, which addresses this point with an additional experiment on ImageNet-100.

---

### Official Review · Reviewer_Ddae · 2023-11-01

**Soundness:** 2 fair
**Presentation:** 3 good
**Contribution:** 3 good
**Rating:** 5
**Confidence:** 4

**Summary:**

This paper studies multi-step classification in the context of Vision Transformers (ViTs). The authors claim that while CNNs are able to effectively learn multi-step classification tasks, ViTs struggle to do the same. The paper investigates this problem in ViTs and discusses the Eureka effect - a sudden improvement in the training and validation performance since the ViT learns the intermediate task. Further, the authors state that this inability to effectively learn the intermediate task stems from the Softmax operation in the self-attention mechanism and present methods to address this issue. Experiments show that the solutions improve the results of the multi-step tasks and ensure faster convergence.

**Strengths:**

Motivation - The paper explores a relatively underexplored problem in the context of ViTs. Specifically, the paper addresses multi-step classification without any supervision for the intermediate task, which is an interesting setting.

Analysis - The paper provides a detailed discussion of why ViTs fail to learn simple multi-step tasks and how the optimization problems can be alleviated. The analysis of the changes in the gradients, attention maps, and the linear probing accuracies of the indicators provide some insights into why ViTs fail in this setting and how Eureka moments are observed.

**Weaknesses:**

(a) Theoretical Concerns:
The problem setting of the paper, the experiments, and the analytical insights are all based on the synthetic MNIST - Fashion MNIST multi-step task. The significance of the whole paper and all the results strongly depends on the assumption that the insights from the experiments with the synthetic data translate to multi-step tasks on data distributions encountered in real-world scenarios. Despite the pivotal significance of this assumption, there is no proof for the same. The authors state in Section 1 that they have provided some indication of the validity of the assumption, but this doesn’t seem to be present in the paper. Therefore, they should provide a sound background that demonstrates how the insights from the experiments with synthetic data scale to real-world data.

(b) Problem Setting:

Relationship between the intermediate tasks - It is possible that the Eureka moments and the failure of ViTs in multi-step tasks are affected by the nature of the relationship between the intermediate tasks. The multi-step classification task used in the paper involves two unrelated tasks, which may affect the insights drawn from the experiments. For example, the majority of the analyses are carried out on a multi-step task based on MNIST and Fashion MNIST, which are not directly related. While the authors consider the worst-case scenario where the tasks are unrelated, they should attempt to explore the effect of the relationship between the tasks on Eureka moments.

Soundness of the setting - While the authors consider the important and practical setting of multi-step classification, the nature of the multi-step task explored in the paper might not be indicative of the settings often encountered in real-world use cases. I agree with the authors that there are several challenges involved in attempting such a study on real-world data, as they have outlined in Section 1. However, the task with 2x2 inputs for the multi-step task on small-scale datasets such as MNIST and CIFAR might not provide meaningful insights that extend to large-scale settings.

Relation to Hierarchical Classification - Hierarchical classification [R1] can be considered as a specific case of multi-step classification where the intermediate tasks are highly related, i.e., the first task involves coarse categories while the second task involves fine categories under each coarse class. The authors should discuss the relationship between hierarchical classification and their proposed multi-step classification setting.

[R1] Miranda, Fábio M., Niklas Köhnecke, and Bernhard Y. Renard. "Hiclass: a python library for local hierarchical classification compatible with scikit-learn." Journal of Machine Learning Research 24.29 (2023): 1-17.

(b) Experiments:
Scale of the models used in the experiments - Table 3 shows that the experiments are conducted on ViT architectures that are similar in scale to ViT-S. However, there are no experiments with larger models, which in part, is limited by the choice of dataset for the multi-step classification task. How do the insights from the smaller models translate to larger architectures such as ViT-B or ViT-L? What is the trend of Eureka moments observed in the larger models?

Experiments with different relationships between the tasks - As mentioned in the previous section, the paper deals with multi-step classification where the intermediate tasks are unrelated. Do the insights on Eureka moments and the optimization challenges presented in the paper hold when the tasks are related? For example, the authors can conduct experiments using the hierarchy of CIFAR-100 or ImageNet (or a subset of ImageNet).

Nature of the multi-step tasks - The current MNIST/CIFAR-based setup does not make a strong case for the study of multi-step classification. The authors should experiment with more representative datasets such as CIFAR-100 and ImageNet. The classes from these datasets provide a wide range of possibilities for the multi-step tasks, and would also allow the authors to analyze the effect of the nature of the intermediate tasks on the occurrence of the Eureka moments.

**Questions:**

How does the proposed multi-step setup relate to hierarchical classification? Can hierarchical classification be considered a specific variant? How would the insights from the current setup translate to a hierarchical setup, if the latter is a specific variant of the former?

How can one study Eureka moments for larger ViT models, since it is likely that they might overfit the proposed MNIST/CIFAR-based classification tasks?

---

> ### Author Response · Authors · 2023-11-20
> **Answer to Ddae**
>
> Thank you for your detailed evaluation and insightful feedback. We address your concerns as listed in the weaknesses below using the same numbering as in your review. We answered the questions below and in the general comment to all reviewers.
>
> **(a)** We address the concerns about relevance and real world transfer in multiple ways. First, we provide additional experiments on realistic high-resolution data, as described in detail in the reply to all reviewers. In addition, we already provided the RoBERTa experiment in the submission (see Fig. 7(b)), where Eureka-Moments are observed on real data and show that the most promising mitigation i.e. NormSoftmax helps also in this case.
> We agree that this is not a theoretical proof that our findings translate to a real world setting, but, we provide sufficient empirical evidence for it.
>
> Please note that we do show an indication for Eureka-Moments on real data and the transfer to real world problems in the paper, as promised in the introduction. We suspect that the RoBERTa experiment was overlooked or not identified as this indication. We admit that the RoBERTa experiment should be highlighted more and in particular these results should be referenced directly in the introduction, where we promise the indication of real world Eureka-Moments and transfer of our findings to the real world. We will modify the final version accordingly.
>
> **(b) Relationship between intermediate tasks.** Thank you for this great suggestion. Indeed, the relation of the tasks is a super interesting direction. Is the task becoming harder or easier when tasks are related? What if similar features need to be used but the tasks are different? All these questions are quite interesting, but unfortunately very difficult to answer in a single paper. How can we measure the relatedness of tasks? Are the tasks we study even completely unrelated or do they not use similar features?
> When training on very related tasks, it opens the door to learn partial solutions to task 1. These small Eureka-Moments will be hard to spot, as the improvements get obscured by task 2 learning. All these additional factors make a clean study and interpretation of the results difficult and would introduce lots of complexity to the paper. We decided to stay in the simple setting for an initial study of Eureka-Moments and left more specific research for future work. Besides this, we already show the RoBERTa experiment in the paper, where tasks are most likely related.
>
> **Another point raised by Ddae is that the relation to hierarchical classification is not discussed in enough detail.** We are happy to add a small discussion to the related works or add it in the introduction. Indeed, we mention hierarchical-classification briefly in our introduction as an example, where Eureka-Moments may also appear. However, this setting is not very well suited to study Eureka-Moments. A flat classification approach, as defined in [R1] does not necessarily have to solve the task via multi-step classification. To give an example: to distinguish cats from dogs the classifier does not necessarily have to classify both as quadrupeds and then classify the leaf nodes. Simply learning a single feature that distinguishes the two is sufficient. Since this alternative strategy exists, we do not know which one is chosen and the choice may even differ on a sample level. Because of these reasons hierarchical-classification as in [R1] is not the right task to study Eureka-Moments for the moment. Nevertheless, hierarchical-classification might benefit from the insights on Eureka-Moments.
>
> **(b) Small scale of models and dataset.** The influence of model scale on Eureka-Moments is important and interesting. As already mentioned by Ddae, we studied the influence of model scale in the supplemental material (see section A.5 and Table 3) and reference this section briefly in the main manuscript. Regarding the concern of dataset size, we would like to refer to the general comment to all reviewers, where we show that large-scale datasets can be easily constructed to study Eureka-Moments in larger networks.
>
> [R1] Miranda, Fábio M., Niklas Köhnecke, and Bernhard Y. Renard. "Hiclass: a python library for local hierarchical classification compatible with scikit-learn." Journal of Machine Learning Research 24.29 (2023): 1-17.
>
> ### Questions
>
> Question regarding hierarchical-classification: We addressed the relation to hierarchical-classification in the section above. Another important distinction is that intermediate tasks are learned implicitly in our setting, but labels for non-leaf nodes are given in general hierarchical-classification. We do not expect Eureka-Moments to appear when direct supervision in the form of labels is available for all sub-tasks.
>
> The last question about studying Eureka moments for larger ViTs has been addressed in the general comment to all reviewers.

---

> ### Comment · Reviewer_Ddae · 2023-12-04
> **Rebuttal Response**
>
> Thanks for the authors’ response and their efforts during the rebuttal period. Even though my concerns about the relevance of the setup and real-world transfer have been addressed, I feel that the following points still deserve attention and can significantly improve the paper:
>
> 1. I feel that the paper is still lacking with respect to the relationship between the intermediate tasks. I agree with the authors that there are numerous possibilities for the choice of intermediate tasks. However, the analysis of Eureka moments is incomplete without it. As the authors have mentioned, the Eureka moments may be observed to a lesser extent or may be nonexistent when the tasks are highly similar. If this is the case, Eureka moments are limited only to highly dissimilar tasks. I believe that the paper can be significantly improved by presenting well-thought experiments for 2-3 different levels of similarity between the intermediate tasks.
> 2. I agree with Reviewer Kjsr that the setting of 2x2 inputs for observing Eureka moments seems somewhat contrived. Do Eureka moments arise only in such 2x2 setups? Even though the authors have shown results on the RoBERTa task that doesn’t exactly follow this setup, the primary analyses and insights from the paper are based on the 2x2 multi-task setup. The authors should present results on a more natural multi-task setup in addition to the setup in the paper.

---

### Official Review · Reviewer_ULYQ · 2023-11-01

**Soundness:** 3 good
**Presentation:** 3 good
**Contribution:** 3 good
**Rating:** 6
**Confidence:** 4

**Summary:**

Paper studies transformers and their ability to understand multi-step decision tasks. This paper mainly compares ViT and ResNet for two multi-step tasks. The work does not only show the deficiencies in a transformer to understand the task, but also reason them and provide a solution. In this work author studies the sudden ability of transformers to learn the subtasks. Mostly, the problem is found to be the Softmax layer in the attention, which leads to small gradients and thusly concludes local uniform attention being the cause for a transformer’s learning problem. The paper describes the creation of two synthetic 2-step task datasets and uses them to compare the performance of ViT and a ResNet. The paper also finds the root cause and tries multiple solutions to remedy the issue. Various ViT model are compared.

**Strengths:**

[1] The paper is clear, well thought out, and detailed.

[2] The author has interesting finding in the transformer in compared to the convolution model. The finding and solution are promising and may have wide applicability.

[3] The dataset creation and explanation were good, the detailed findings of the shortcomings of a transformer and the explanation of the solutions were well supported by the set of experiments done.

[4] The extensive analysis presented in this paper regarding the issue is both thorough and impressive.

**Weaknesses:**

[1] The experiment is carefully designed and demonstrates issues where the transformer architecture lags behind the convolutional architecture. However, the experiments are conducted on a small-scale dataset (MNIST/CIFAR), which may not be very representative of real-world scenarios. The real-world scenario may be much more challenging. I kindly request the author to provide results over the tinyImageNet or ImageNet100 datasets.

[2] It will be interesting to see the robustness of the proposed solution, i.e., how it behaves when faced with more than two multi-step decision tasks. Also, If we extend the model beyond just the vision domain, for instance, in a multimodal setting, do the same assumptions and proposed solutions still hold?

[3] Reproducing the results may be challenging. I kindly request the author to please provide the code for replication.

**Questions:**

Please refer to the weakness section.

---

> ### Author Response · Authors · 2023-11-20
> **Answer to ULYQ**
>
> We would like to thank you for your time spent on this review and thoughtful suggestions and questions. We address your concerns in the general response to all reviewers and below, using the same numbering as in your review.
>
>  1) Please see the general comment to all reviewers for our response to the point raised regarding small-scale dataset and transferability to real-world scenarios.
>
>  2) We are not entirely sure we understand the request regarding the robustness. We would like to emphasize that we do not claim that NormSoftmax is a general one-size-fits-all solution. Instead, we use NormSoftmax as a minimal intervention of the Softmax function to show that the problems is related to the Attention block. Regarding robustness, we also conduct experiments on the harder "No Position Task" and find that NormSoftmax fails for 1 out of 4 training runs.
>
>
>    - The second part of the question addresses 3 or more step tasks. While never tested explicitly, we have no evidence that 3 step tasks behave fundamentally different. We suspect the Eureka-Ratio to be much smaller. While definitely interesting to investigate the behavior for tasks with more than 2 sub-tasks, we believe that it complicates studying the phenomenon drastically and might obscure other findings in the paper. In case you have a particular question or experiment in mind that helps in understanding the phenomenon using 3 or more-step tasks, we would be very grateful for your input.
>
>
>   -  Regarding the last part of the question whether the phenomenon appears beyond the visual domain, we definitely think that this is the case and this is an interesting future research direction. We do already provide evidence that this happens for BERT training in the paper. Furthermore, we show with the "reasoning task" (see Fig. 7(a)), that we observe the same even for a single layer transformer with "pre-extracted features". This experiment simulates for example the setup of two frozen feature extractors (potentially multi-modal) that are "fused" using a transformer block.
>
> 3) We agree that reproducibility is extremely important and the code will be released the latest upon acceptance as mentioned in our paper. We strongly believe that providing code alongside a paper is important.

---

### Official Review · Reviewer_Kjsr · 2023-11-04

**Soundness:** 3 good
**Presentation:** 2 fair
**Contribution:** 2 fair
**Rating:** 5
**Confidence:** 4

**Summary:**

This paper introduces a setting in which the authors discover what's termed a "eureka moment" where both training and validation loss decrease quickly after initially both saturating.  The setting they propose consists of a two-stage task where the transformer has to learn first by comparing top left and bottom right squares to see whether it needs to classify the image in the top right or bottom left.  They show transformers often struggle with this two-stage task while ResNets do not and hypothesize poor gradient flow through the softmax resulting from either attention collapse or too disperse attention weights. They propose adding temperature or using NormSoftmax to alleviate small gradient norms and show increased frequency of Eureka moments than standard attention without adding temperature.  These approaches have higher rate of eureka moments than using standard Softmax attention.

**Strengths:**

- As far as I know, Eureka moment is a new category of phase transition phenomenon where both training and validation curves saturate before then suddenly making progress.  This differs from Grokking where training is saturated by not validation.
- The authors identify mitigations that increase the rate of reaching Eureka moments for a simple ViT architecture.  At a high level, these mitigations add entropy to attention softmax and include using a fixed temperature, or an increasing schedule for temperature (termed by the authors Heat Treatment), or a previously proposed NormSoftmax that uses the minimum of the empirical standard deviation of inputs and temperature.

**Weaknesses:**

- Setting seems contrived and signal for Eureka moments is much weaker for Roberta model compared to ViT.  The two-stage tasks studied is structure such that the model needs to be able to learn the relationship between task 1 and task 2 and not become hyper-fixated on the classification task in 2.  The proposed mitigations directly address the structure by effectively encouraging exploration during training so that the relationship can be discovered.
- Adding entropy to softmax increases rate of Eureka moments but there isn't sufficient evidence for low gradient norm being the cause when Eureka moments do not occur.  In particular, Figure 5 doesn't show a big difference between norms of Vit vs NormSoftmax based on what I can tell with log axis.
- The authors do not sufficiently identify the source of the problem.  In particular, I expected a deeper investigation into the role of initialization; missing Eureka moments seems like it could be driven by bad initialization which is then fixed with higher entropy.

Typos:
- "With other initializations they never learn the __taks__ 1 within 1000 epochs"
- Unusual to see artefact instead of artifact.

**Questions:**

- How do the L1 Gradient Norms look for ViT without temperature in an extended horizon through reaching eureka moment?  How does this compare to the norms for a run of ViT without Eureka moment?
- What is the role of initialization in this?  Can we address the issue with better initialization schemes instead?  My sense is yes since the mitigation schemes effectively encourage exploration throughout the training process to ensure the model can eventually learn the relationship between task 1 and task 2.
- For roberta in Figure 7B, is the loss curve training or validation?  How do the training and validation curves compare for this task?
- Does a Eureka moment exist for larger ViT architectures with more layers?

---

> ### Author Response · Authors · 2023-11-20
> **Answer to Kjsr**
>
> ## Answer to Kjsr
>
> Thank you for your review. We highly acknowledge your thoughtful review and valuable comments. We address your point in the general comment to all reviewers and below.
>
> **Differences of gradient norms.** One point raised by the reviewer Kjsr is that the difference between the gradient norms in Figure 5 is not big. We would like to point out that initially the difference between W_V gradients, and W_K and W_Q gradient norms is more than an order of magnitude for ViT. After about 1200 steps it settles at about 0.5 orders of magnitudes. For NormSoftmax, on the other hand, there is very little difference. Differences only become apparent after the Eureka-Moment (indicated by the end of the gray box in Figure 5).
>
> **Gradient norms for ViTs**. The L1 gradient norms for ViT with and without Eureka-Moment can hardly be distinguished. Only briefly before the Eureka-Moment it can be observed that gradient norms increase sharply. Note, that this is not in contrast with our explanation.
> Gradient updates are small for W_K and W_Q for all vanilla ViTs. As correctly identified by you, random initialization also plays a big role. Depending on the random initialization of the weights the required amount of change of the weights is different. With a small gradient norm for W_K and W_Q many small steps will be needed  to reach a Eureka-Moment. If the gradient norms are larger, fewer steps will be needed. This is in particular in line with our results when training for 3000 epochs instead of 300. In this setting all ViTs have a Eureka-Moment.
>
>
>
> **Role of initialization**: To test the role of the initialization we implemented the initialization described in [1]. Initialization may help prevent getting into such a "bad energy landscape", where parts of the gradients become very small or initialization might just initialize the network weights closer to a Eureka-Moment. Alternatively, initialization may kick-start the learning process but could lead to the same problems eventually.
> To ensure a fair comparison, we followed exactly the same hyper-parameter search and training as described in the main paper. We find that the Eureka-Ratio of ViTs using the initialization described in [1] is similar to the one observed for normal ViTs. To be precise the Eureka-Ratio for our standard dataset (Table 1 left) is 7/10 with a mean accuracy of 89.31 and 0/4 Eureka-Ratio for the "No Position Task". Given that these results are very similar to the vanilla ViT we conclude that this type of initialization does not seem to resolve the problem.
> We suspect that despite the superior initialization the early training steps lead to equivalently bad energy landscapes with small gradients for W_Q and W_K and equally "far away".
> We will extend Table 1 with this baseline for the final version of the paper.
>
> **RoBERTa loss curve**: In Figure 7(b) we show the *training* loss, as this is the relevant metric to study and also distinguishes it from grokking. The validation curve looks similar, but was not included since we ran evaluation very coarsely during training. We will repeat the training with more frequent evaluation and add training and validation loss curves to the supplemental material. We will also change the y-axis label in Figure 7(b) to training loss. Thank you for pointing this out.
>
> **Eureka-Moments in deeper ViTs**: Depth is one of the parameters we tested in Table 3 (supplemental material). We still observe Eureka-Moments for deeper networks. In addition to these results, we added experiments with ViT-S on ImageNet-100. Please refer to the general comment to all reviewers for these experiments.
>
> Thanks for pointing out the typos. We will revise the manuscript accordingly.
>
> ### References
> [1] Trockman, Asher, and J. Zico Kolter. "Mimetic Initialization of Self-Attention Layers." arXiv 2023.

---

> > ### Author Response · Authors · 2023-11-22
> > **RoBERTa loss curve added to supplemental material**
> >
> > We just added the figure showing both, training and validation loss for the RoBERTa experiments to the supplemental material (Figure 22).

---

> ### Comment · Reviewer_Kjsr · 2023-11-23
> **Post author response**
>
> I appreciate the additional experiments the authors provided for a more challenging task as well as trying a different initialization scheme. I will raise my score accordingly. However, I do still think the task is somewhat contrived to benefit from during training and think the work could benefit from additional examples of eureka moment.

---

### Author Response · Authors · 2023-11-20
**General answer to all reviewers**

## General Answer

We would like to thank all reviewers for their valuable feedback, suggestions and constructive critique. We are very grateful to receive such detailed reviews.

In this general comment we will address requests, questions, and critiques that multiple reviewers brought up. The remaining, more specific questions will be addressed in comments to the respective reviews.

## Do Eureka-Moments happen for high-resolution, large realistic datasets and larger models?

One common point of critique of reviewers was that we study Eureka-Moments using MNIST and fashion-MNIST with transformers for small scale datasets. This experimental design led to concerns whether our results transfer to real settings or are just an artifact of MNIST-like data.

To further address these points we create a large-scale dataset with high-resolution natural images using ImageNet-100. For simplicity, we follow the same dataset design as for the MNIST-like datasets, i.e. we place the targets in the top-right and bottom left, while the other two quadrants show indicator images. Targets are simply images from ImageNet-100. The indicators are sampled from 2 of the ImageNet dog classes. If both indicators show the exact same sample, the top-right image needs to be classified and bottom left otherwise. The probability of top-right location being the target is 0.5.
We train "ViT-S" and "ViT-S with NormSoftmax" following the standard ImageNet training setting using strong augmentations from Deit training (https://github.com/facebookresearch/deit/blob/main/README_deit.md). Vanilla ViT obtains only ~41% accuracy and we don't observe a Eureka-Moment. For ViT-S with NormSoftmax we observe a Eureka-Moment at epoch 131. The training and validation curves are shown in the updated supplemental material in Figure 21 (last figure).

This experiment addresses multiple reviewer questions and requests at once:
1) It addresses the concern that small-scale datasets might not be representative for the real world (@ULYQ, @Ddae, @Lp2g).
2) The observation of Eureka-Moments on real, high-resolution data and the success of NormSoftmax shows that both problem as well as analysis transfer to realistic realistic settings (@Ddae, @Lp2g).
3) It shows that both, problem as well as analysis transfer to large-scale settings (@Ddae).
4) For this experiment we train ViT-S, which is significantly larger than the ViTs trained on the MNIST-like dataset. Thus, we show that Eureka-Moments happen for larger architectures with more layers. (also see Table. 3 in the supplemental material for an analysis of model scale, i.e. heads, embedding dimension and layers) (@Kjsr).
5) We show that insights from small architectures transfer to larger architectures and show how to study Eureka-Moments for larger ViT models. Please note that larger architectures beyond ViT-S are out of scope for us given the time (@Ddae).

Despite these new results, we would like to direct the attention also to the RoBERTa experiments in Figure 7(b) of the main paper. It shows a Eureka-Moment on real text data. Furthermore, the same intervention used for the MNIST-like experiments (i.e. NormSoftmax) leads to a faster Eureka-Moment, indicating that our analysis on MNIST-like synthetic data transfers to realistic settings (even in other modalities) and has practical utility.

---

### Meta-Review · Area_Chair_t1Y1 · 2023-12-06

**Metareview:**

This paper investigates a unique training phenomenon in two-stage transformer tasks, referred to as the "Eureka moment," characterized by a rapid decrease in both training and validation loss after initial stagnation. The study attempts to identify the cause of this phenomenon as the softmax function and proposes strategies to mitigate it. Reviewers commend the contributions but express concerns about the artificial nature of the task setups, the simplicity of the datasets employed, and the limited exploration of various intermediate task categories. During the rebuttal phase, the authors made a good effort to clarify these concerns. However, reviewers maintain reservations regarding the suitability of the demonstrated tasks and experimental configurations, particularly the designed 2x2 input tasks, for generalizing observations of Eureka moments to more practical scenarios. This paper could benefit from more comprehensive experimental designs, with a specific focus on real-world tasks and intermediate tasks exhibiting varying degrees of similarity.

**Justification For Why Not Higher Score:**

The paper did not receive a higher score primarily because of concerns about the generalizability of the "Eureka moment" findings. Reviewers pointed out issues with the artificiality of task setups, the simplicity of datasets, and the limited range of intermediate task categories explored. Despite efforts to address these in the rebuttal, the paper's experimental design, particularly the 2x2 input tasks, was seen as insufficient for extrapolating the observed phenomenon to more practical scenarios.

**Justification For Why Not Lower Score:**

N/A

---

### Decision · Program_Chairs · 2024-01-16

Reject